# Connecting the Patches: Multivariate Long-term Forecasting using Graph and Recurrent Neural Network

## Abstract

Many Transformer-based models have achieved great performance on multivariate long-term time series forecasting (MLTSF) tasks in the past few years, but they are ineffective in capturing cross-channel dependencies and temporal order information. In multivariate time series analysis, the cross-channel dependencies can help the model understand the correlations between multivariate time series, and the consistency of time series is also essential for more accurate predictions. Therefore, we propose GRformer, adopting the Graph neural network (GNN) and position encoding based on recurrent neural network (RNN) to better process multivariate time series data. We design a mix-hop propagation layer and embed it in the feedforward neural network to encourage proper interaction between different time series. To introduce temporal order information, we use a multi-layer RNN to recursively generate positional embeddings for sequence elements. Experiments on eight real-world datasets show that our model can achieve more accurate predictions on MLTSF tasks.

## 1 Introduction

Time series forecasting is an indispensable part of many fields, such as traffic flow forecasting (Guo et al., 2019), energy management (Uremović et al., 2022), weather forecasting (Zhang et al., 2022), and finance (Sezer et al., 2020), etc.

In the past few years, Transformer-based models have achieved great success in various fields, such as natural language processing (NLP) (Kalyan et al., 2021), computer vision (CV) (Han et al., 2022), etc. This trend extends to multivariate long-term time series forecasting (MLTSF) tasks (Wen et al., 2022). Models like Informer (Zhou et al., 2021), Autoformer (Wu et al., 2021), and FEDformer (Zhou et al., 2022a) make better predictions than previous works (Lai et al., 2018; Bai et al., 2018) based on recurrent neural network (RNN) or convolutional neural network (CNN). Most of these Transformer models adopt the channel-mixing strategy. They extract features from all sequence values at each time step and use the self-attention mechanism (Vaswani et al., 2017) to capture long-term temporal dependencies globally. However, the effectiveness of these Transformer-based models has been challenged by DLinear (Zeng et al., 2023), which introduces a channel-independent strategy and makes more accurate predictions with a simple linear model. Recently, an improved Transformer-based model PatchTST (Nie et al., 2023) is proposed. PatchTST applies the attention mechanism independently to patches so that the distribution of attention weights varies across different channels, thus enabling the model to distinguish time series with different behaviors.

However, various correlations may exist between different channels of multivariate time series, as shown in Figure 1(a), and this could be an important factor in multivariate time series forecasting. Current Transformer-based models are ineffective in capturing the cross-channel dependencies of the time series. For channel-mixing Transformer models, the weight matrices of Multilayer Perceptron (MLP) layers of the feedforward network in Encoder and Decoder multiply with the input embeddings, which implicitly capture the channel dependencies of different time series. But such operation may result in chaotic information interactions, as shown in Figure 1(b). For channel-independent models, recent studies emphasize the individuality of different time series while ignoring the commonalities between different channels.

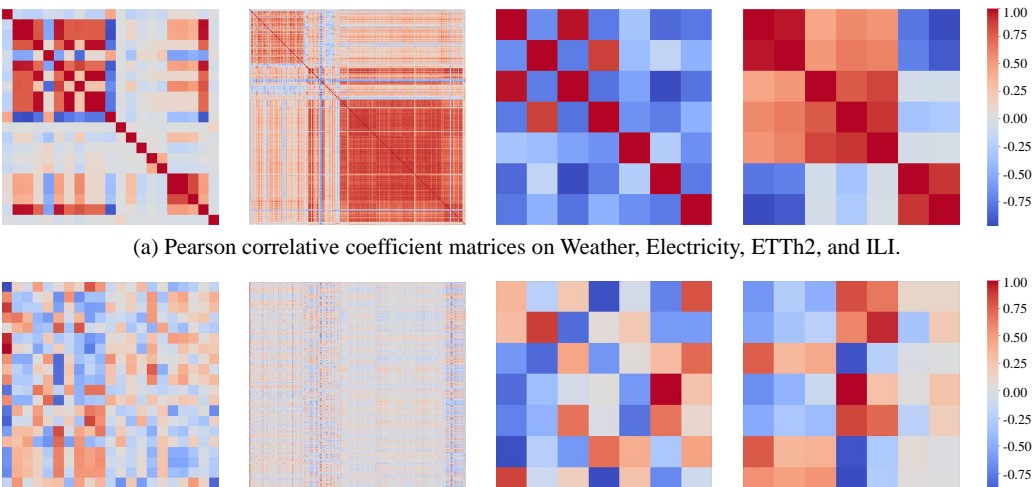

(a) Pearson correlative coefficient matrices on Weather, Electricity, ETTh2, and ILI.

(b) MLP weight matrices product results of previous channel-mixing Transformer-based models on Weather, Electricity, ETTh2, and ILI. The weight matrices' parameters are obtained from Informer.

Figure 1: The correlation heat maps of four datasets: Weather, Electricity, Electricity Transformer Temperature-hourly (we choose ETTh2), Influenza-Like Illness (ILI). The number of channels in Weather, Electricity, ETTh2, ILI are 21, 321, 7 and 7 respectively. (a) shows the linear correlation between different channels. (b) is the equivalent correlation matrices obtained by multiplying the weight matrices of the MLP layers of embedding projectors and the feedforward network in Encoders and Decoders of Informer (Zhou et al., 2021). The data distribution is highly dispersed, implicitly resulting in chaotic information interactions.

Furthermore, time series analysis is based on continuous data points (Zeng et al., 2023). The strict order of the sequence elements is important because temporal order is a crucial factor in determining the trend, periodicity, and other characteristics of time series data. Most of the present Transformer-based models use fixed or learnable position encoding methods (Vaswani et al., 2017; Devlin et al., 2018; Liu et al., 2019) to inject position-wise information to sequence tokens. Although these methods can help models understand positional variance, they don't explicitly consider strict temporal order information and have the problem of information loss and redundancy when facing long-term time series (Dehghani et al., 2019).

Therefore, we propose GRformer, an improved Transformer-based model for MLTSF tasks. To capture the correlations among the channels of multivariate time series, we design a graph convolutional module, motivated by (Wu et al., 2020). The module consists of a mix-hop propagation layer that can make different channels focusing on their multi-hop neighbors. To help the model identify the importance of neighborhoods' information of different hops, we parameterize a matrix to assign different weights to them. To obtain an adjacency matrix that correctly represents the correlations between different channels, we use the Pearson correlation coefficient algorithm (Cohen et al., 2009) and a filter function. To capture temporal order information, we use a multi-layer RNN to generate positional embeddings. The sequential modeling capability of RNN helps to generate positional encodings with temporal order information, thus enabling the model to make use of the strict order information of the time series. The main contributions of this paper are:

- We propose a Pearson coefficient-based graph constructing module and use an improved mix-hop propagation layer to capture the cross-channel dependencies of time series so that each channel can focus on others that are highly correlated with it. We embed this layer in the feedforward network to integrate graph neural network with Transformer.
- We design a multi-layer RNN structure for position encoding so that the representations at different time steps can carry strict temporal order information.
- We conduct extensive experiments on eight commonly used datasets. Our model ranks first in performance on seven of these datasets and achieves performance improvement with a 5.7% decrease in Mean Square Error (MSE) and a 6.1% decrease in Mean Absolute Error (MAE) compared to the current state-of-the-art (SOTA) model.

## 2 RELATED WORK

Transformer-based models perform well in modeling long-term temporal dependencies with their self-attention mechanism. However, the quadratic complexity to the length of a sequence limits these models' application on long-term time series forecasting and leads to researches on this issue. LogTrans (Li et al., 2019) proposes *LogSparse* attention to solve locality-agnostics and memory bottlenecks. Reformer (Kitaev et al., 2020) replaces the dot-product attention with a locality-sensitive hashing attention. Informer (Zhou et al., 2021) proposes a self-attention distillation operation called *ProbSparse*, which selects top-$k$ elements of the attention weight matrix based on KL divergence. Autoformer (Wu et al., 2021) adopts a temporal decomposition approach and introduces a self-correlation mechanism to replace self-attention. Pyraformer (Liu et al., 2022b) introduces a pyramidal attention module to summarize features at different resolutions and model the temporal dependencies. FEDFormer (Zhou et al., 2022a) learns from Fourier decomposition and applies Transformer in the frequency domain rather than the temporal domain.

Most of the above models adopt the point-wise attention and channel-mixing strategy. Specifically, elements of different sequences at each time step are first mapped to a feature representation and then fed into the encoder for further processing. However, the same attention weight will be assigned to elements at the same time step. This approach ignores the various behaviors of multivariate time series, and the MLP layers in the feedforward neural network may further cause chaotic interactions among time series.

Recently, some channel-independent models have been proposed. These models emphasize processing each channel of the multivariate time series independently. DLinear (Zeng et al., 2023) utilizes a simple linear layer to convert each channel directly to the output sequence. It challenges the effectiveness of previous Transformer models with an outstanding prediction accuracy. PatchTST (Nie et al., 2023) borrows the ideas of Dlinear and Vision Transformer (Dosovitskiy et al., 2021), dividing the sequence of each channel into patches and using a patch-wise attention. This approach can enrich the semantics of extracted features and distinguish the various behaviors of different channels. However, these models ignore the correlations between different channels of multivariate time series. Crossformer (Zhang & Yan, 2023) proposes a two-stage attention approach. It uses a Cross-Time attention layer to process patches like PatchTST, and uses a Cross-Dimension attention layer to make patches from different channels interact, thus capturing the correlations between different channels. However, the use of decoder and the routing mechanism in the Cross-Dimension attention layer limits its predictive performance (Zhou et al., 2022b).

## 3 METHODOLOGY

### 3.1 PROBLEM DEFINITION

We consider the MLTSF task as follows: given a multivariate time series $\mathbf{X} = \left\{x_{(1,t)}, x_{(2,t)}, ..., x_{(M,t)}\right\}_{t=1}^{L}$, we predict the future values $\mathbf{Y} = \left\{y_{(1,t)}, y_{(2,t)}, ..., y_{(M,t)}\right\}_{t=L+1}^{L+\tau}$, where $L$ refers to the length of the look-back window, $M$ represents the number of sequences, and $\tau$ is the prediction horizons.

We use graph structure to capture correlations between different time series. Graph is used to describe the relationships between different entities within a network, and it can be represented as $\mathcal{G} = (\mathcal{V}, \mathcal{E})$, where $\mathcal{V}$ is the set of nodes and $\mathcal{E}$ is the set of edges. We abstract individual time series as nodes, that is, $|\mathcal{V}| = M$. We use an adjacency matrix $\mathbf{A} \in \mathbb{R}^{|\mathcal{V}| \times |\mathcal{V}|}$ to store the graph where $\mathbf{A}_{i,j} = c > 0$ if there is a high-intensity correlation between two time series $x_{(i)}$ and $x_{(j)}$ $(i \neq j)$, otherwise, $\mathbf{A}_{i,j} = 0$.

### 3.2 MODEL STRUCTURE

The overall architecture of GRformer is shown in Figure 2(a). We capture the temporal and channel dependencies independently by extracting temporal order information and encouraging different channels to interact with each other. For temporal dependencies, we introduce a RNN-based position encoding method to help our model understand temporal order information. For channel dependencies, we first initialize an adjacency matrix using the Pearson correlation coefficient algorithm and

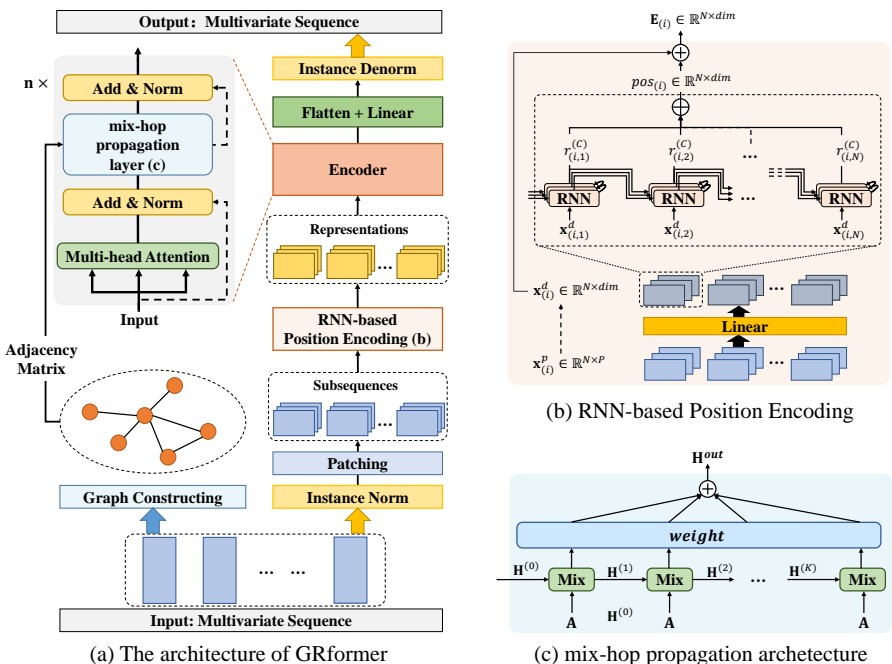

Figure 2: The architecture of GRformer. (a) Each channel of multivariate time series is divided into patches. The multi-layer RNN injects temporal order information and the mix-hop propagation layer captures cross-channel dependencies. (b) The representations of patches of a channel are passed through a multi-layer RNN to generate positional embeddings. The hidden state of the $j$-th unit in the $(c-1)$-th layer is passed to the $j$-th unit in the $c$-th layer and the $(j+1)$-th unit in the $(c-1)$-th layer. (c) The mix-hop module aggregates the information of multi-hop neighbors.

then use a graph convolutional module embedded in the feedforward network to aggeregate the information from different channels. The module consists of a mix-hop propagation layer, which can effectively capture the relationships between different time series channels. Finally, we get outputs through a Flatten layer and Feature Head because a recent research (Zhou et al., 2022b) argues that the Transformer decoder may cause model performance degradation in long-term time series forecasting tasks.

### 3.2.1 RNN-BASED POSITION ENCODING

Previous transformer-based models represent elements at different positions in the sequence by injecting positional variance information. However, unlike other sequence data, time series data has strong temporal continuity. In order for the model to make use of the strict temporal order of the time series, we consider using a multi-layer RNN to inject enhanced positional contextual information into the sequence tokens.

Each channel univariate sequence $x_{(i)}$ is first divided into patches $x_{(i)}^p \in \mathbb{R}^{N \times P}$ and then passed through a linear layer to generate representations $x_{(i)}^d \in \mathbb{R}^{N \times dim}$. The variable $P$ is the length of each patch, $S$ refers to the stride between different patches, and $N$ is the number of patches which is calculated by $N = \lfloor \frac{L-P}{S} + 1 \rfloor$. Each representation is fed into the corresponding RNN unit, as shown in Figure 2(b). We concatenate the hidden states of the last layer to get positional embeddings. The detailed progress is defined in Eq 1:

$$
\begin{aligned}
r_{(i,j)}^{(c)} &= \text{RNN}(r_{(i,j-1)}^{(c)}, r_{(i,j)}^{(c-1)}), \ c \in \{1, 2, ..., C\} \\
pos_{(i)} &= \text{Concat}(r_{(i,1)}^{(C)}, r_{(i,2)}^{(C)}, ..., r_{(i,N)}^{(C)}) \\
\mathbf{E}_{(i)} &= x_{(i)}^d + pos_{(i)}
\end{aligned}
\tag{1}
$$

where $\text{RNN}(\cdot)$ could be any kinds of RNN model, such as RNN, LSTM and GRU. $C$ denotes the number of RNN layers. $x_{(i,j)}^d$ represents the $j$-th patch representation of the $i$-th channel. $r_{(i,j)}^{(c)}$ is the hidden state obtained from the corresponding $j$-th RNN unit in the $c$-th layer and $r_{(i,j)}^{(0)} = x_{(i,j)}^d$. The variable $pos_{(i)}$ and $\mathbf{E}_{(i)}$ are the positional embeddings and final representations of the $i$-th channel respectively. The obtained positional embeddings ensure the positional variance of each time step and introduce strict temporal order information by recursively generating embeddings at later positions using the preceding ones. The RNN structure does not directly determine the final embeddings output, so the gradient vanishing problem is alleviated to some extent.

### 3.2.2 GRAPH CONSTRUCTING

Before the training progress, we first generate a graph structure based on Pearson correlation coefficient algorithm. For a time series channel pair $(x_{(i)}, x_{(j)})$, where $i \neq j$, we calculate the Pearson correlation coefficient between them. The results are saved in a correlation matrix $\mathbf{A}^{Pearson} \in \mathbb{R}^{M \times M}$, where $\mathbf{A}_{i,j}^{Pearson} = \mathbf{A}_{j,i}^{Pearson} = \rho_{x_{(i)}, x_{(j)}} = \frac{\text{conv}(x_{(i)}, x_{(j)})}{\sigma_{x_{(i)}} \sigma_{x_{(j)}}}$. The function $\text{conv}(x_{(i)}, x_{(j)})$ is the covariance of $(x_{(i)}, x_{(j)})$ and $\sigma$ is the standard deviation. We use a filter function $f$ and retain top-k elements to get the adjacency matrix $\mathbf{A}$. The definition is as follows:

$$f(x) = \begin{cases} x, & \text{if } x > \lambda \\ 0, & \text{if } x \leq \lambda \end{cases} \tag{2}$$

$$\mathbf{A} = \text{argtopk}(f(\mathbf{A}^{Pearson})) \tag{3}$$

where the function $\text{argtopk}(\cdot)$ returns the top-k largest values of each row in a matrix, $\lambda$ is a threshold hyperparameter to filter the correlation of different time series.

Our graph structure ensures that relationships are established for those highly correlated time series channels. Some recent works (Wu et al., 2020; Liu et al., 2022a) adopt adaptive graph learning layers to generate a graph. However, these approaches ignore the inherent relationships between time series and bring extra computation complexity of $O(M^2)$. Our method is independent of the training process and does not have extra complexity.

### 3.2.3 MIX-HOP PROPAGATION

We embed a mix-hop propagation layer into the feedforward neural network to aggregate the information of a channel with its related neighbors, motivated by (Wu et al., 2020). As depicted in Figure 2(c), we first propagate multi-hop neighbor information in the graph structure, defined as follows:

$$\begin{aligned} \mathbf{H}_{i,j}^{(k)} &= \alpha \mathbf{H}_{i,j}^{(0)} + (1 - \alpha)\tilde{\mathbf{A}}\mathbf{H}_{i,j}^{(k-1)} \\ \tilde{\mathbf{A}} &= \tilde{\mathbf{D}}^{-1}(\mathbf{A} + \mathbf{I}) \\ \tilde{\mathbf{D}}_{ii} &= 1 + \sum\nolimits_{j=1}^{N} \mathbf{A}_{ij} \end{aligned} \tag{4}$$

where $\mathbf{H}^{(0)}$ is each node's original state outputted by the preceding multi-head attention layer. To avoid the nodes' hidden states converging to a single point, we use $(\mathbf{A} + \mathbf{I})$ to establish the self-correlation of a node and a hyperparameter $\alpha$ to retain a proportion of nodes' original states during the propagation process. $\tilde{\mathbf{D}}$ is the degree matrix of $(\mathbf{A} + \mathbf{I})$ and $\tilde{\mathbf{A}}$ is the normalized adjacency matrix. We use a weighted matrix to aggregate the results of different depths of propagation. The process is defined as Eq 5:

$$\begin{aligned} weight &= \text{Softmax}(Parameter(a_1, a_2, ..., a_K)) \\ \mathbf{H}_{i,j}^{out} &= \sum\nolimits_{k=1}^{K} \mathbf{H}_{i,j}^{(k)} \cdot weight_k \end{aligned} \tag{5}$$

where $K$ represents the depth of message propagation and $weight$ is a learnable parameter containing $K$ variables randomly initialized. We use the Softmax function to assign different weights to the multi-hop neighbor information. We sum the results and get the final output $\mathbf{H}_{i,j}^{out}$.

### 3.2.4 Loss function

We use MAE Loss, which measures the average absolute discrepancy between predicted values and ground truth. The definition of the Loss function is as follows:

$$\mathcal{L}(Y, \hat{Y}) = \frac{1}{|Y|} \sum_{i=1}^{|Y|} |y_{(i)} - \hat{y}_{(i)}|. \tag{6}$$

where $Y$ is the ground truth, and $\hat{Y}$ represents the predicted values.

### 3.2.5 Instance Normalization

The statistical properties of time series usually change over time, resulting in a changing data distribution. Time series in the real world usually conform to the characteristics of non-stationary sequences(Du et al., 2021). Therefore, we consider using an instance normalization technique called RevIN (Kim et al., 2022) to alleviate the distribution drift problem. RevIN eliminates non-stationary statistics in the input sequence and improves the robustness of the model by normalizing the input sequences and denormalizing the output of the model. The instance normalization process corresponds to the **Instance Norm** module and the **Instance Denorm** module in Figure 2(a).

## 4 Experiments

### 4.1 Experimental settings

**Datasets.** We evaluate our model on 8 real-world datasets: Weather, Electricity, Traffic, 4 ETT datasets(ETTh1, ETTh2, ETTm1, ETTm2), and ILI. These datasets are extensively utilized, covering multiple fields including weather, energy, transportation, and healthcare. The statistics of the datasets are shown in Table 1. ETT and ILI have a small number of sequences, and the correlations between different channels are simple. Weather, Electricity, and Traffic have a large number of sequences, and the cross-channel dependencies in these datasets are complex. More detailed information of the datasets is in Appendix A.1.1.

Table 1: Statistics of popular datasets for benchmark.

| Dataset | Metrics | Frequency | Length |
|---|---|---|---|
| Weather | 21 | 10 min | 52696 |
| Electricity | 321 | 1 hour | 26304 |
| Traffic | 862 | 1 hour | 17544 |
| ETTh1 | 7 | 1 hour | 17420 |
| ETTh2 | 7 | 1 hour | 17420 |
| ETTm1 | 7 | 15 min | 69680 |
| ETTm2 | 7 | 15 min | 69680 |
| ILI | 7 | 7 day | 966 |

**Baselines.** We select the following Transformer-based models and non-Transformer-based models including PatchTST (Nie et al., 2023), Crossformer (Zhang & Yan, 2023), FEDformer (Zhou et al., 2022a), Autoformer (Wu et al., 2021), and DLinear (Zeng et al., 2023). Full information and descriptions of the baselines can be found in Appendix A.1.2.

**Experimental Settings.** We follow the settings in (Nie et al., 2023) and use different look-back windows $L$ and future predicting horizons $\tau$. For GRformer, PatchTST, Crossformer, and Dlinear, we set $L = 104$ and $\tau \in \{24, 36, 48, 60\}$ for ILI that has a very small amount of data, while $L = 336$ and $\tau \in \{96, 192, 336, 720\}$ for other datasets. For other Transformer-based models, we use the default look-back window $L = 96$. Considering that PatchTST and Crossformer use the patching technique as GRformer, we set $P = 24$ and $S = 2$ on ILI while $P = 16$ and $S = 8$ on other datasets for all three models. We evaluate GRformer with $C \in \{1, 2, 3\}$, $\lambda \in \{0.8, 0.6, 0.4, 0.2\}$ and $\alpha \in \{0.03, 0.05, 0.1, 0.15\}$ and choose the best results. For the selection of metrics to evaluate the performance, we utilize MSE and MAE. More detailed settings can be found in Appendix A.1.3 and A.1.4.

**Results.** Table 2 shows the results of multivariate long-term forecasting. Overall, our model achieves better performance than other baselines on all datasets with different prediction lengths. Specifically, compared with the current SOTA Transformer-based model PatchTST, the MSE is reduced by 4.06% and the MAE is reduced by 5.08% on average. The improvement of MSE and MAE reaches to 12.94% and 12.78% on average compared with DLinear, the current best non-Transformer-based model for MLTSF. To evaluate the influence of the loss function, we test our

model using MSE Loss, MAE Loss (the default loss function), and Huber Loss. GRformer achieves optimal average performance when using the MAE loss. MSE Loss takes the square of the prediction error, which is more sensitive to outliers. This result indicates that our model is more robust to outliers. Overall, GRformer shows excellent performance in the cases of both normal conditions and frequent outliers. The detailed definition and results of different loss functions can be found in Appendix A.3. The Crossformer can also capture cross-channel dependencies between time series. However, the predictive performance of Crossformer is not very stable, which may be related to its utilization of the decoder and the unstable routing mechanism. Compared to Crossformer, our model is additionally capable of capturing temporal order information, resulting in the better performance. We also conduct experiments on univariate time series forecasting, the detailed results can be found in Appendix A.2.

Table 2: Experimental results of MLTSF task on 8 real-world datasets. Models with ∗ follow the experimental results from the original papers and PatchTST. The best results are in **bold** and the second best results are underlined.

| Models | | GRformer | | PatchTST | | DLinear | | FEDformer* | | Autoformer* | | Crossformer | |
|---|---|---|---|---|---|---|---|---|---|---|---|---|---|---|
| Metric | | MSE | MAE | MSE | MAE | MSE | MAE | MSE | MAE | MSE | MAE | MSE | MAE |
| Weather | 96 | **0.147** | **0.184** | 0.152 | 0.200 | 0.176 | 0.226 | 0.238 | 0.314 | 0.249 | 0.329 | 0.156 | 0.218 |
| | 192 | **0.192** | **0.232** | 0.196 | 0.237 | 0.219 | 0.261 | 0.275 | 0.329 | 0.325 | 0.370 | 0.198 | 0.262 |
| | 336 | **0.245** | **0.273** | 0.249 | 0.283 | 0.266 | 0.296 | 0.339 | 0.377 | 0.351 | 0.391 | 0.266 | 0.295 |
| | 720 | **0.318** | **0.325** | 0.321 | 0.334 | 0.333 | 0.342 | 0.389 | 0.409 | 0.415 | 0.426 | 0.327 | 0.363 |
| Traffic | 96 | **0.363** | **0.224** | 0.367 | 0.251 | 0.410 | 0.282 | 0.576 | 0.359 | 0.597 | 0.371 | 0.503 | 0.281 |
| | 192 | **0.385** | **0.232** | **0.385** | 0.259 | 0.423 | 0.287 | 0.610 | 0.380 | 0.607 | 0.382 | 0.537 | 0.329 |
| | 336 | **0.394** | **0.239** | 0.398 | 0.265 | 0.436 | 0.296 | 0.608 | 0.375 | 0.623 | 0.387 | 0.552 | 0.356 |
| | 720 | **0.433** | **0.258** | 0.434 | 0.287 | 0.466 | 0.315 | 0.621 | 0.375 | 0.639 | 0.395 | 0.598 | 0.377 |
| Electricity | 96 | **0.126** | **0.217** | 0.130 | 0.222 | 0.140 | 0.237 | 0.186 | 0.302 | 0.196 | 0.313 | 0.137 | 0.238 |
| | 192 | **0.142** | **0.234** | 0.147 | 0.240 | 0.153 | 0.249 | 0.197 | 0.311 | 0.211 | 0.324 | 0.159 | 0.268 |
| | 336 | **0.155** | **0.248** | 0.165 | 0.258 | 0.169 | 0.267 | 0.213 | 0.328 | 0.214 | 0.327 | 0.173 | 0.286 |
| | 720 | **0.184** | **0.276** | 0.202 | 0.291 | 0.203 | 0.301 | 0.233 | 0.344 | 0.236 | 0.342 | 0.210 | 0.304 |
| ETTh1 | 96 | **0.365** | **0.387** | 0.371 | 0.397 | 0.375 | 0.397 | 0.376 | 0.415 | 0.435 | 0.446 | 0.402 | 0.418 |
| | 192 | **0.406** | **0.412** | 0.412 | 0.422 | 0.413 | 0.420 | 0.423 | 0.446 | 0.456 | 0.457 | 0.469 | 0.458 |
| | 336 | **0.430** | **0.429** | 0.437 | 0.437 | 0.439 | 0.443 | 0.441 | 0.462 | 0.486 | 0.487 | 0.588 | 0.540 |
| | 720 | **0.429** | **0.452** | 0.451 | 0.466 | 0.479 | 0.493 | 0.469 | 0.492 | 0.515 | 0.517 | 0.725 | 0.610 |
| ETTh2 | 96 | **0.274** | **0.332** | **0.274** | 0.336 | 0.289 | 0.353 | 0.332 | 0.374 | 0.332 | 0.368 | 0.706 | 0.587 |
| | 192 | **0.337** | **0.373** | 0.338 | 0.376 | 0.383 | 0.418 | 0.407 | 0.446 | 0.426 | 0.434 | 0.855 | 0.689 |
| | 336 | **0.355** | **0.390** | 0.356 | 0.397 | 0.448 | 0.465 | 0.400 | 0.447 | 0.477 | 0.479 | 1.013 | 0.767 |
| | 720 | **0.382** | **0.417** | 0.385 | 0.425 | 0.605 | 0.551 | 0.412 | 0.469 | 0.453 | 0.490 | 1.131 | 0.800 |
| ETTm1 | 96 | **0.281** | **0.326** | 0.290 | 0.342 | 0.299 | 0.343 | 0.326 | 0.390 | 0.510 | 0.492 | 0.299 | 0.353 |
| | 192 | **0.325** | **0.356** | 0.344 | 0.387 | 0.336 | 0.364 | 0.365 | 0.415 | 0.514 | 0.495 | 0.344 | 0.387 |
| | 336 | **0.357** | **0.377** | 0.366 | 0.392 | 0.369 | 0.386 | 0.392 | 0.425 | 0.510 | 0.492 | 0.421 | 0.438 |
| | 720 | **0.417** | **0.413** | 0.419 | 0.425 | 0.425 | 0.421 | 0.446 | 0.458 | 0.527 | 0.493 | 0.562 | 0.524 |
| ETTm2 | 96 | **0.161** | **0.246** | 0.166 | 0.254 | 0.167 | 0.260 | 0.180 | 0.271 | 0.205 | 0.293 | 0.269 | 0.362 |
| | 192 | **0.213** | **0.284** | 0.222 | 0.294 | 0.224 | 0.303 | 0.252 | 0.318 | 0.278 | 0.336 | 0.462 | 0.463 |
| | 336 | **0.266** | **0.319** | 0.277 | 0.330 | 0.281 | 0.342 | 0.324 | 0.364 | 0.343 | 0.379 | 0.741 | 0.600 |
| | 720 | **0.351** | **0.372** | 0.366 | 0.386 | 0.397 | 0.421 | 0.410 | 0.420 | 0.414 | 0.419 | 1.160 | 0.792 |
| ILI | 24 | **1.281** | **0.731** | 1.422 | 0.789 | 2.215 | 1.081 | 2.624 | 1.095 | 2.906 | 1.182 | 3.537 | 1.215 |
| | 36 | **1.104** | **0.671** | 1.497 | 0.847 | 1.963 | 0.963 | 2.516 | 1.021 | 2.585 | 1.038 | 3.559 | 1.222 |
| | 48 | **1.255** | **0.718** | 1.438 | 0.813 | 2.130 | 1.024 | 2.505 | 1.041 | 3.024 | 1.145 | 3.776 | 1.257 |
| | 60 | **1.314** | **0.737** | 1.530 | 0.868 | 2.368 | 1.096 | 2.742 | 1.122 | 2.761 | 1.114 | 3.932 | 1.285 |

## 4.2 ABLATION STUDY

For position encoding, we adopt three different approaches to generate positional embeddings, namely fixed position encoding, learnable position encoding, and RNN-based position encoding. We use MSE as the loss function for these three methods. For capturing the correlation of time series, we design two sets of experimental conditions: one using the mix-hop propagation layer while the other not. We adopt MAE loss for these two conditions using GNN. We test our model with different RNN layers $C$, Pearson coefficient filter threshold $\lambda$, top-k number and propagation ratio $\alpha$ and choose the best results. All results are in Appendix A.5.1, A.5.2, A.5.3 and A.5.4.

As shown in Table 3, compared to learnable position encoding(i.e. PatchTST), RNN-based position encoding achieved reductions of 2.51% and 1.83% in MSE and MAE on average, while the graph convolutional module achieves 0.65% and 3.63%. Although the fixed position encoding can make different time steps distinguishable, its fixed embedding values do not contain much temporal order information. The learnable position encoding implicitly learns temporal order information to some extent and performs better compared to the fixed strategy. Our RNN-based position encoding adopts an explicit structure and can provide positional embeddings with more specific temporal order information, thus enhancing the model's ability to adapt to time series data.

The results indicate that injecting temporal order information and make highly correlated time series focus on each other's features to obtain similar trends, periodicity, etc. can help to make more accurate predictions.

Table 3: Ablation study of RNN-based position encoding and channel mixing strategy. There are five cases listed: (a) only use RNN to generate all position embeddings (R); (b) only use learnable position encoding, which is PatchTST (L); (c) only use fixed position encoding (F); (d) only use graph convolution module (Mix-CN); (e) use both RNN-based position encoding and graph convolution module, which is our GRformer (R & Mix-CN). The best results are in **bold** and second best results are underlined.

| Models | | R | | L | | F | | Mix-CN | | R & Mix-CN | |
|---|---|---|---|---|---|---|---|---|---|---|---|
| Metric | | MSE | MAE | MSE | MAE | MSE | MAE | MSE | MAE | MSE | MAE |
| Weather | 96 | 0.149 | 0.187 | 0.152 | 0.200 | 0.152 | 0.204 | **0.145** | **0.184** | _0.147_ | **0.184** |
| | 192 | _0.193_ | 0.236 | 0.196 | 0.239 | 0.198 | 0.249 | 0.195 | **0.230** | **0.192** | _0.232_ |
| | 336 | **0.235** | **0.267** | 0.249 | 0.283 | 0.258 | 0.287 | 0.247 | _0.273_ | _0.245_ | _0.273_ |
| | 720 | **0.316** | 0.330 | 0.321 | 0.334 | 0.330 | 0.341 | 0.322 | _0.327_ | _0.318_ | **0.325** |
| Traffic | 96 | **0.356** | 0.241 | 0.367 | 0.251 | 0.370 | 0.255 | 0.382 | _0.232_ | _0.363_ | **0.224** |
| | 192 | **0.378** | 0.251 | _0.385_ | 0.259 | 0.388 | 0.265 | 0.395 | _0.239_ | _0.385_ | **0.232** |
| | 336 | **0.392** | 0.260 | 0.398 | 0.265 | 0.402 | 0.275 | 0.407 | _0.249_ | _0.394_ | **0.239** |
| | 720 | **0.426** | 0.281 | _0.436_ | 0.287 | 0.441 | 0.292 | 0.451 | _0.274_ | 0.443 | **0.258** |
| Electricity | 96 | _0.129_ | 0.223 | 0.130 | 0.223 | 0.132 | 0.226 | 0.130 | _0.219_ | **0.126** | **0.217** |
| | 192 | _0.146_ | 0.239 | 0.148 | 0.240 | 0.149 | 0.243 | 0.146 | _0.236_ | **0.142** | **0.234** |
| | 336 | 0.163 | 0.257 | 0.165 | 0.258 | 0.168 | 0.260 | _0.162_ | _0.253_ | **0.155** | **0.248** |
| | 720 | 0.200 | 0.289 | 0.202 | 0.291 | 0.211 | 0.295 | _0.199_ | _0.283_ | **0.188** | **0.276** |
| ETTh1 | 96 | 0.370 | 0.394 | 0.371 | 0.396 | 0.372 | 0.396 | _0.366_ | _0.388_ | **0.365** | **0.387** |
| | 192 | 0.409 | 0.418 | 0.412 | 0.422 | 0.412 | 0.429 | _0.407_ | _0.413_ | **0.404** | **0.410** |
| | 336 | 0.432 | 0.434 | 0.438 | 0.436 | 0.442 | 0.439 | **0.428** | **0.425** | _0.430_ | _0.429_ |
| | 720 | 0.447 | 0.458 | 0.451 | 0.466 | 0.452 | 0.467 | _0.439_ | _0.459_ | **0.429** | **0.452** |
| ETTh2 | 96 | 0.277 | 0.338 | **0.274** | 0.336 | 0.282 | 0.346 | 0.276 | _0.333_ | **0.274** | **0.332** |
| | 192 | 0.343 | 0.379 | _0.339_ | 0.379 | 0.350 | 0.388 | 0.342 | **0.374** | **0.338** | **0.374** |
| | 336 | 0.360 | 0.397 | _0.356_ | 0.397 | 0.362 | 0.403 | 0.364 | _0.394_ | **0.355** | **0.390** |
| | 720 | 0.387 | 0.423 | _0.385_ | 0.425 | 0.400 | 0.436 | 0.394 | _0.419_ | **0.382** | **0.417** |
| ETTm1 | 96 | 0.285 | 0.340 | 0.290 | 0.342 | 0.301 | 0.351 | 0.291 | 0.331 | **0.281** | **0.326** |
| | 192 | 0.326 | 0.367 | 0.344 | 0.387 | 0.346 | 0.388 | 0.332 | 0.359 | **0.325** | **0.356** |
| | 336 | 0.361 | 0.391 | 0.366 | 0.392 | 0.380 | 0.408 | 0.367 | 0.381 | **0.357** | **0.377** |
| | 720 | 0.413 | 0.421 | 0.419 | 0.425 | 0.421 | 0.424 | 0.421 | 0.418 | **0.417** | **0.413** |
| ETTm2 | 96 | 0.164 | 0.255 | 0.166 | 0.254 | 0.171 | 0.256 | **0.161** | **0.247** | **0.161** | **0.247** |
| | 192 | 0.220 | 0.293 | 0.222 | 0.294 | 0.230 | 0.294 | _0.216_ | _0.285_ | **0.213** | **0.284** |
| | 336 | 0.272 | 0.328 | 0.277 | 0.331 | 0.272 | 0.328 | _0.268_ | **0.319** | **0.266** | **0.319** |
| | 720 | 0.360 | 0.385 | 0.366 | 0.386 | 0.371 | 0.393 | _0.356_ | _0.375_ | **0.351** | **0.372** |
| ILI | 24 | _1.303_ | 0.753 | 1.422 | 0.789 | 1.611 | 0.856 | 1.433 | _0.748_ | **1.281** | **0.731** |
| | 36 | _1.328_ | 0.772 | 1.497 | 0.847 | 1.585 | 0.799 | 1.462 | _0.767_ | **1.104** | **0.671** |
| | 48 | **1.250** | 0.767 | 1.438 | 0.813 | 1.629 | 0.837 | 1.491 | 0.802 | _1.255_ | **0.718** |
| | 60 | _1.447_ | _0.823_ | 1.530 | 0.868 | 1.685 | 0.876 | 1.513 | 0.826 | **1.314** | **0.737** |

## 4.3 Varying look-back window

To see the effect of different sizes of look-back window, we conduct experiments with $L \in \{96, 192, 336, 720\}$ and $\tau \in \{96, 192, 336, 720\}$, and the results on three datasets are shown in Figure 3. It can be found that as the look-back window size increases, Autoformer tends to over-fit. Crossformer makes significant improvement, but its overall performance is biased and severely affected by $L$, while GRformer achieves the best results and stable performance improvement. All results of our model can be found in Appendix A.7.

Table 4: The computational complexity of different models. $L$ represents the length of the historical look-back window, $M$ is the number of time series, $P$ denotes the patch length, $C$ is the number of RNN layers, $\tau$ is the future prediction horizons, $dim$ is the dimension of the latent state, and $K$ is the depth of propagation.

| Models | Position Encoding | Encoder layer | Decoder layer |
|---|---|---|---|
| Autoformer (Wu et al., 2021) | $O(1)$ | $O(L \log L)$ | $O((\frac{L}{2} + \tau) \log(\frac{L}{2} + \tau))$ |
| FEDformer (Zhou et al., 2022a) | $O(1)$ | $O(L)$ | $O(\tau + \frac{L}{2})$ |
| Crossformer (Zhang & Yan, 2023) | $O(\frac{L}{P})$ | $O(M(\frac{L}{P})^2)$ | $O(M^{\frac{\tau(\tau+L)}{P^2}})$ |
| PatchTST (Nie et al., 2023) | $O(\frac{L}{P})$ | $O(M(\frac{L}{P})^2)$ | $O(M^{\frac{\tau L dim}{P}})$ |
| GRformer (Ours) | $O(C\frac{L}{P})$ | $O(KM(\frac{L}{P})^2)$ | $O(M^{\frac{\tau L dim}{P}})$ |

### 4.4 COMPLEXITY ANALYSIS

We analyze the computational complexity of the position encoding process, encoder, and decoder of different models, the results are shown in Table 4. The computational complexity of simple single-layer recurrent networks such as vanilla RNNs, LSTMs, and GRUs is linear with the length of the input sequence. The patching operation reduces the number of sequence elements from $L$ to $\frac{L}{P}$. Therefore, the multi-layer RNN in GRformer has the computational complexity of $O(C\frac{L}{P})$. Because of the channel-independent strategy, the attention weights need to be calculated for $M$ times, so there is $O(M(\frac{L}{P})^2)$ computational complexity of one attention layer. Additionally, the gradients need to be aggregated from $K$-hop neighbors, resulting in the $O(KM(\frac{L}{P})^2)$ complexity of the encoder layer. The Graph construction operation is independent of the training process, and the mix-hop propagation layer is equivalent to matrix multiplication without gradient update, so there isn't extra computational complexity added. We replace the computational complexity of the Decoder layer with that of Flatten layer with linear head because we use this module to get outputs. The complexity is $O(M\frac{\tau L dim}{P})$, which is the same as PatchTST.

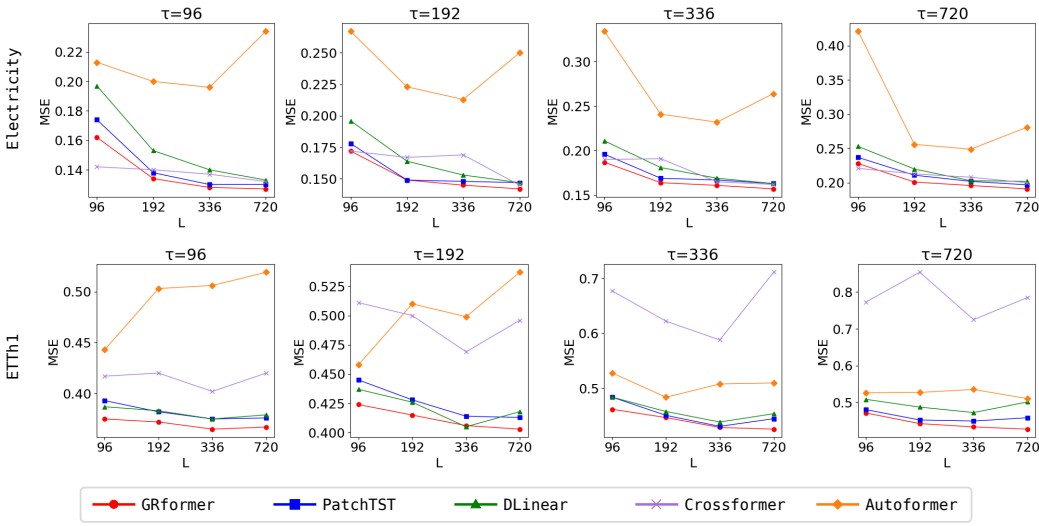

Figure 3: The results of MSE using varying look-back window size $L$ on three datasets: Electricity, ETTh1, and ETTm2. The look-back windows $L$ is selected from $\{96, 192, 336, 720\}$, and the prediction horizons $\tau \in \{96, 192, 336, 720\}$. A total of five models are listed: PatchTST, DLinear, Autoformer, Crossformer, and our GRformer.

## 5 CONCLUSION

We propose GRformer, an improved Transformer-based MLTSF model with two key designs: a mix-hop propagation layer in the feedforward network and RNN-based position encoding. Compared to previous works, our mix-hop propagation layer demonstrates the effectiveness of capturing cross-channel dependencies in multivariate time series forecasting tasks. To ensure proper correlations between different channels, we take advantage of the Pearson correlation coefficient to generate a reasonable graph structure. The RNN-based position encoding we adopt can not only distinguish position difference information but also ensure that the obtained position embeddings have temporal order information. This factor can't be captured by previous MLTSF models.

There are still some questions that need to be answered. Although the RNN structure has been used to optimize the position encoding process(Wang et al., 2019), there is still no strict theory or indicator to measure the temporal order characteristic exhibited by the extracted features. What's more, the universality of MLTSF models in short-term forecasting and more practical application scenarios still needs to be verified, such as cloud-edge workload forecasting (Wang et al., 2022) in complex IoT business scenarios which contains more randomness. Future researches need to be done to explore more diverse and complex scenarios.

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

# A Appendix

## A.1 Experimental Details

### A.1.1 Datasets

We test our model on the following 8 real-world multivariate datasets following (Wu et al., 2021):

1. **Weather**[1] contains 21 meteorological indicators for Germany in 2020, such as air pressure, temperature, and humidity etc. The time interval for data records is 10 minutes.

2. **Traffic**[2] records the road occupancy rates from 862 sensors on San Francisco Bay area freeways from July 2016 to July 2018. The time frequency is one hour.

3. **Electricity**[3] contains 321 customers' hourly electricity consumption from July 2016 to July 2019.

4. **ETT**[4] (Electricity Transformer Temperature) contains 7 indicators of an electricity transformer in two years, the dataset is divided to 1 and 2 according to the number of machines, and based on the time interval between data records (1h and 15min), it is further marked as h and m, namely ETTh1, ETTh2, ETTm1 and ETTm2.

5. **ILI**[5] (Influenza-Like Illness) records the number of patients in different age groups in a weekly frequency.

Note that for **Weather**, **Traffic**, **Electricity** and **ILI**, the ratio we split to train/validation/test set is 0.7/0.1/0.2. For **ETTh1**, **ETTh2**, **ETTm1**, and **ETTm2**, the train/validation/test set contains 12/4/4 months of data respectively.

### A.1.2 Baselines

The brief decription of our baseline methods is as follows:

1. **Autoformer** (Wu et al., 2021) uses series decomposition technique with Auto-Correlation mechanism to capture cross-time dependency for long-term time series forecasting.

2. **FEDformer** (Zhou et al., 2022a) utilizes trend decomposition and Fourier transformation techniques and use Transformer to process frequency domain features. It is the model with best performance before DLinear.

3. **DLinear** (Zeng et al., 2023) decomposes time series into two different components, and generates a single Linear layer for each of them. Such a simple design has defeated all the complex transformer models proposed before it.

4. **Crossformer** (Zhang & Yan, 2023) is aware of the fact that segmenting subsequences in LSTF is beneficial. Different from the current SOTA, it reintroduce channel mixing technique to serve its purpose of multivariate time series forecasting. To capture channel dependencies between time series, it designs another attention layer, working with a routing mechanism to reduce complexity.

5. **PatchTST** (Nie et al., 2023) adopts patching and channel independent techniques, making semantic extraction of time series from a single time step to multiple time steps, and achieve SOTA in LSTF models.

### A.1.3 Model Parameters

By default, we set hyperparameters shown in Table 6. The backbone of GRformer contains 3 encoder layers, with hidden state dimension $dim$ as 128 and multi-attention heads as 8. For the mix-hop layer in feed forward network, the depth of propagation $K$ is set to 2, while the two linear layers in the

---

[1]https://www.bgc-jena.mpg.de/wetter/

[2]https://pems.dot.ca.gov/

[3]https://archive.ics.uci.edu/ml/datasets/ElectricityLoadDiagrams20112014

[4]https://github.com/zhouhaoyi/ETDataset

[5]https://gis.cdc.gov/grasp/fluview/fluportaldashboard.html

feedforward network would transform the features from $dim = 128$ to $d\_ff = 256$ and then back to $dim = 128$. Finally, the results will go through an GELU (Hendrycks & Gimpel, 2016) activation function. Note that for datasets with less time series, namely ETTh1, ETTh2, ETTm1, ETTm2 and ILI, we will only keep top-2 elements of Pearson correlation coefficient matrix to generate the adjacency matrix $\mathbf{A}$.

We use the same historical look-back window $L$ as 336 and future horizon steps $T \in \{96, 192, 336, 720\}$ for GRformer, PatchTST, Crossformer and DLinear, to make a relatively fair comparison. Note that for Crossformer, the original datasets, length of look-back window and future horizens it uses are not the same as ours, but based on the fact that all the three models of Crossformer, PatchTST and GRformer adopt Patching technique and are adapted to MLSTF task, it is reasonable to keep the input and output lengths the same for these models. For dataset like ILI that has few instances, we set $L$ to 104 and do prediction on $T \in \{24, 36, 48, 60\}$.

For different datasets, we use $optuna$ (Akiba et al., 2019) to retrieve the optimal parameters. We tune a total of seven parameters, namely learning rate, dropout, RNN depth $C$, filter threshold $\lambda$, top-k number, propagation depth $K$, and the propagation ratio $\alpha$. The learning rate has the greatest impact on the results, which is the main parameter of fine-tuning. Other parameters cause slight differences in the final results. The searching space is shown in Table 5. The learning rate used in different datasets are shown in Table 6. We adopt Adam optimizer to adjust model parameters for minimizing the loss function.

Table 5: The searching space of learning rate, dropout, RNN depth $C$, filter threshold $\lambda$, top-k number, propagation depth $K$, and the propagation ratio $\alpha$.

| Hyperparameter | Searching Space | |
| --- | --- | --- |
| Learning rate | $\{6.0e-06, 1.0e-05, 2.0e-05, 3.0e-05, 8.0e-05, 1.0e-04, 2.0e-04, 3.0e-04, 8.0e-04, 1.0e-03, 1.2e-03, 2.0e-03\}$ | |
| dropout | $\{0.15, 0.2, 0.25, 0.3, 0.35, 0.4\}$ | |
| $C$ | $\{1, 2, 3\}$ | |
| $\lambda$ | $\{0.8, 0.6, 0.4, 0.2\}$ | |
| top-k | ETT & ILI | $\{1, 2, 3\}$ |
| | Weather | $\{2, 4, 8\}$ |
| | Electricity & Traffic | $\{4, 8, 16, 32\}$ |
| $\alpha$ | $\{0.03, 0.05, 0.1, 0.15\}$ | |

Table 6: The learning rate on different datasets. We use scientific notation to display the numerical value of the learning rate.

| Datasets | Weather | Traffic | Electricity | ETTh1 | ETTh2 | ETTm1 | ETTm2 | ILI |
| --- | --- | --- | --- | --- | --- | --- | --- | --- |
| Learning rate | $8.0e-05$ | $8.0e-04$ | $2.0e-04$ | $2.0e-05$ | $1.0e-05$ | $3.0e-05$ | $1.0e-05$ | $1.0e-03$ |
| $C$ | 1 | 2 | 1 | 1 | 1 | 2 | 2 | 1 |
| $\lambda$ | 0.4 | 0.4 | 0.6 | 0.8 | 0.8 | 0.8 | 0.8 | 0.8 |
| top-k | 4 | 16 | 8 | 3 | 3 | 3 | 3 | 2 |
| $\alpha$ | 0.1 | 0.05 | 0.15 | 0.05 | 0.1 | 0.05 | 0.1 | 0.05 |

As for the other baseline models, we will keep the same strategy as experiments in PatchTST.

### A.1.4 ENVIRONMENTS

We run our experiments on Ubuntu 18.04 LTS Linux system, using cuda 11.8, Pytorch 2.0.0 and Python 3.8. Due to the large scale of data magnitude variation, we choose different hardware facilities for the experiment. For datasets like Weather, ETT, and ILI that have limited time series, we select one RTX 3060-Ti GPU with 8GB of memory. For Electricity and Traffic, which have a very large number of time series, we select two V100 GPUs, each of which has 32GB memory.

### A.2 UNIVARIATE FORECASTING

We evaluate our model's performance on univariate forecasting tasks using the ETT datasets. We try to predict the specific variable "Oil Temperature (OT)". The results of different models are shown in Table 7.

Table 7: Univariate forecasting results on ETT datasets with the future prediction horizons $\tau \in \{96, 192, 336, 720\}$. The best results are in **bold**.

| Models | | GRformer | | PatchTST | | DLinear | | FEDformer | | Autoformer | | Crossformer | |
|---|---|---|---|---|---|---|---|---|---|---|---|---|---|
| Metric | | MSE | MAE | MSE | MAE | MSE | MAE | MSE | MAE | MSE | MAE | MSE | MAE |
| ETTh1 | 96 | 0.056 | **0.179** | **0.055** | **0.179** | 0.056 | 0.180 | 0.079 | 0.215 | 0.071 | 0.206 | 0.062 | 0.190 |
| | 192 | **0.071** | 0.205 | **0.071** | 0.215 | **0.071** | **0.204** | 0.104 | 0.245 | 0.114 | 0.262 | **0.071** | 0.208 |
| | 336 | 0.087 | 0.235 | 0.085 | 0.232 | 0.098 | 0.244 | 0.119 | 0.270 | 0.107 | 0.258 | **0.076** | **0.214** |
| | 720 | 0.099 | 0.249 | **0.087** | **0.232** | 0.189 | 0.359 | 0.142 | 0.299 | 0.126 | 0.283 | 0.098 | 0.247 |
| ETTh2 | 96 | **0.126** | **0.277** | 0.129 | 0.282 | 0.131 | 0.279 | 0.128 | 0.271 | 0.153 | 0.306 | 0.196 | 0.351 |
| | 192 | **0.165** | **0.322** | 0.168 | 0.328 | 0.176 | 0.329 | 0.185 | 0.330 | 0.204 | 0.351 | 0.257 | 0.411 |
| | 336 | **0.184** | **0.345** | 0.185 | 0.351 | 0.209 | 0.367 | 0.231 | 0.378 | 0.246 | 0.389 | 0.283 | 0.433 |
| | 720 | **0.199** | **0.359** | 0.224 | 0.383 | 0.276 | 0.426 | 0.278 | 0.420 | 0.268 | 0.409 | 0.375 | 0.501 |
| ETTm1 | 96 | **0.026** | 0.122 | **0.026** | **0.121** | 0.028 | 0.123 | 0.033 | 0.140 | 0.056 | 0.183 | 0.084 | 0.239 |
| | 192 | 0.040 | **0.150** | **0.039** | **0.150** | 0.045 | 0.156 | 0.058 | 0.186 | 0.081 | 0.216 | 0.102 | 0.261 |
| | 336 | **0.053** | 0.174 | **0.053** | 0.173 | 0.061 | 0.182 | 0.084 | 0.231 | 0.076 | 0.218 | 0.127 | 0.293 |
| | 720 | **0.073** | **0.203** | 0.074 | 0.207 | 0.080 | 0.210 | 0.102 | 0.250 | 0.110 | 0.267 | 0.121 | 0.279 |
| ETTm2 | 96 | **0.063** | **0.181** | 0.065 | 0.186 | **0.063** | 0.183 | 0.067 | 0.198 | 0.065 | 0.189 | 0.096 | 0.241 |
| | 192 | **0.091** | **0.224** | 0.094 | 0.231 | 0.092 | 0.227 | 0.102 | 0.250 | 0.110 | 0.267 | 0.133 | 0.285 |
| | 336 | **0.119** | **0.261** | 0.120 | 0.265 | **0.119** | **0.261** | 0.130 | 0.279 | 0.154 | 0.305 | 0.169 | 0.326 |
| | 720 | **0.166** | **0.317** | 0.171 | 0.322 | 0.175 | 0.320 | 0.178 | 0.325 | 0.182 | 0.335 | 0.283 | 0.429 |

## A.3 RESULTS OF DIFFERENT LOSS FUNCTIONS

We conduct experiments for GRformer using different loss function, we test our model with Mean Squared Error (MSE) Loss, Mean Absolute Error (MAE) Loss and Huber Loss. The specific definition of these loss functions are in Eq 7, Eq 8, and Eq 9.

$$\text{MSE}(Y, \hat{Y}) = \frac{1}{|Y|} \sum_{i=1}^{|Y|} |Y_i - \hat{Y}_i|^2 \tag{7}$$

$$\text{MAE}(Y, \hat{Y}) = \frac{1}{|Y|} \sum_{i=1}^{|Y|} |Y_i - \hat{Y}_i| \tag{8}$$

$$\text{HUBER}(Y, \hat{Y}) = \begin{cases} \frac{1}{2} \cdot \frac{1}{|Y|} \sum_{i=1}^{|Y|} |Y_i - \hat{Y}_i|^2, & \text{if } |Y_i - \hat{Y}_i| \leq \delta \\ \frac{1}{|Y|} \sum_{i=1}^{|Y|} \delta |Y_i - \hat{Y}_i| - \frac{1}{2} \delta^2, & \text{if } |Y_i - \hat{Y}_i| > \delta \end{cases} \tag{9}$$

We use MAE Loss by default. For Huber Loss, we set $\delta = 1.0$. We conduct experiments on the 8 datasets, the results are shown in Table 8.

## A.4 HOMOSCEDASTICITY VERIFICATION

The correct expression of Pearson correlation coefficient for sequences depends on the homoscedasticity of the data. Before the training progress, we standardize the data of each dataset to avoid the impact of different sequences on model training due to different unit scales. The Pearson coefficient results are obtained using the data with almost identical variance after standardization. We use the 'bartlett' function from the Python 'scipy' library for Bartlett variance analysis, and the results show that the data used in our model conforms to the homoscedasticity hypothesis.

## A.5 HYPERPARAMETER SENSITIVITY ANALYSIS

To verify whether GRformer is sensitive to the hyperparameters, we conduct experiments of varying model parameters. The fine-tuned hyperparameters are: (a) RNN layer depth $C$, (b) Pearson coefficient matrix filter threshold $\lambda$, (c) top-k number for selecting edges in the adjacency matrix and (d) propagation ratio $\alpha$ for aggregating information of different hops. For each setting, we repeat the experiment 5 times with 100 epochs (with early-stop control) each time and report the average MAE with a standard deviation. We change the parameter under investigation and fix other parameters in each experiment. Overall, on different datasets, adjusting the number of hyperparameters does not cause large fluctuations in the results. Different datasets also have their own preferences for dif-

Table 8: Experiments of different Loss functions on 8 datasets. Three cases are listed: (a) is GRformer with MAE Loss, which is the default setting; (b) is GRformer with MSE Loss; (c) is GRformer with Huber Loss. The best results are in **bold**.

| Models | | GRformer(MAE) | | GRformer(MSE) | | GRformer(Huber) | |
|---|---|---|---|---|---|---|---|
| Metric | | MSE | MAE | MSE | MAE | MSE | MAE |
| Weather | 96 | **0.147** | **0.184** | 0.150 | 0.199 | 0.149 | 0.191 |
| | 192 | **0.192** | **0.232** | 0.196 | 0.243 | 0.196 | 0.236 |
| | 336 | **0.245** | **0.273** | 0.250 | 0.286 | 0.248 | 0.279 |
| | 720 | **0.318** | **0.325** | 0.322 | 0.336 | 0.321 | 0.331 |
| Traffic | 96 | 0.363 | **0.234** | **0.354** | 0.239 | 0.363 | 0.236 |
| | 192 | 0.385 | **0.232** | **0.376** | 0.249 | 0.384 | 0.245 |
| | 336 | 0.394 | **0.239** | **0.390** | 0.258 | 0.399 | 0.251 |
| | 720 | 0.433 | **0.258** | **0.424** | 0.280 | 0.434 | 0.275 |
| Electricity | 96 | **0.126** | **0.217** | 0.128 | 0.224 | 0.128 | 0.221 |
| | 192 | **0.142** | **0.234** | 0.145 | 0.240 | 0.146 | 0.238 |
| | 336 | **0.155** | **0.248** | 0.165 | 0.261 | 0.162 | 0.256 |
| | 720 | **0.184** | **0.276** | 0.198 | 0.290 | 0.196 | 0.285 |
| ETTh1 | 96 | **0.365** | **0.387** | 0.372 | 0.395 | 0.370 | 0.390 |
| | 192 | **0.406** | **0.412** | 0.410 | 0.419 | 0.409 | 0.416 |
| | 336 | **0.430** | **0.429** | 0.434 | 0.436 | 0.432 | 0.433 |
| | 720 | **0.429** | **0.452** | 0.446 | 0.465 | 0.437 | 0.459 |
| ETTh2 | 96 | **0.274** | **0.332** | 0.284 | 0.342 | 0.277 | 0.336 |
| | 192 | **0.337** | **0.373** | 0.347 | 0.383 | 0.346 | 0.381 |
| | 336 | **0.355** | **0.390** | 0.372 | 0.403 | 0.360 | 0.393 |
| | 720 | **0.382** | **0.417** | 0.399 | 0.433 | 0.385 | 0.420 |
| ETTm1 | 96 | **0.281** | **0.326** | 0.287 | 0.342 | 0.285 | 0.336 |
| | 192 | **0.325** | **0.356** | 0.332 | 0.370 | 0.327 | 0.363 |
| | 336 | **0.357** | **0.377** | 0.367 | 0.395 | 0.362 | 0.387 |
| | 720 | **0.417** | **0.413** | 0.422 | 0.424 | 0.425 | 0.420 |
| ETTm2 | 96 | **0.161** | **0.246** | 0.165 | 0.256 | **0.161** | 0.249 |
| | 192 | 0.213 | **0.284** | 0.220 | 0.294 | **0.212** | 0.286 |
| | 336 | **0.266** | **0.319** | 0.273 | 0.328 | **0.266** | 0.320 |
| | 720 | 0.351 | 0.372 | 0.359 | 0.383 | **0.349** | **0.371** |
| ILI | 24 | **1.281** | **0.731** | 1.417 | 0.782 | 1.393 | 0.746 |
| | 36 | **1.104** | **0.671** | 1.277 | 0.751 | 1.336 | 0.735 |
| | 48 | **1.255** | 0.718 | 1.283 | 0.759 | 1.286 | **0.710** |
| | 60 | **1.314** | **0.737** | 1.612 | 0.864 | 1.486 | 0.790 |

ferent hyperparameters, this difference is obvious for hyperparameters related to the GNN module, because the relationships between time series vary across different datasets.

### A.5.1 RNN LAYER DEPTH

With the increase of RNN layer depth, more information from previous RNN units can be propagated to the later ones. Considering that it makes it harder for the model to optimize the parameters of RNN that is too deep, we conduct experiments of using RNN layers $C \in \{1, 2, 3\}$. The results can be seen in Figure 4 (a). In most cases, the optimal result is obtained when the number of RNN layers $C$ equals to 1 or 2.

### A.5.2 FILTER THRESHOLD FOR GRAPH CONSTRUCTION

The filter threshold $\lambda$ is used to filter out sequence pairs that are not highly correlated. The smaller the $\lambda$, the more likely the inter-sequence interactions are to be taken into account. A large $\lambda$ may result in insufficient sequence interaction information being used, while a small $\lambda$ may make a sequence referencing more weakly correlated neighbors and introduce too much noise. The empirical choice is that when the Pearson correlation coefficient is greater than 0.8, the two sequences can be considered to be strongly positively correlated. However, in order to comprehensively consider the interaction information between time series, we conduct the threshold selection experiments. We set the top-k value to be the same as the number of sequences in each dataset to see the full impact of the threshold. The results corresponding to different $\lambda$ is shown in Figure 4 (b). In most cases, it is better for datasets that have large number of sequences to consider more interactions. However, if $\lambda$ is set to 0.2, all the cases get the worst results, which indicate that a small threshold take too many weakly correlated neighbors into account that too much noise is aggregated by one sequence. The best threshold for Traffic, Electricity, ETTh1 and ETTm2 is $\lambda = 0.4$, $\lambda = 0.6$, $\lambda = 0.8$ and $\lambda = 0.8$ respectively.

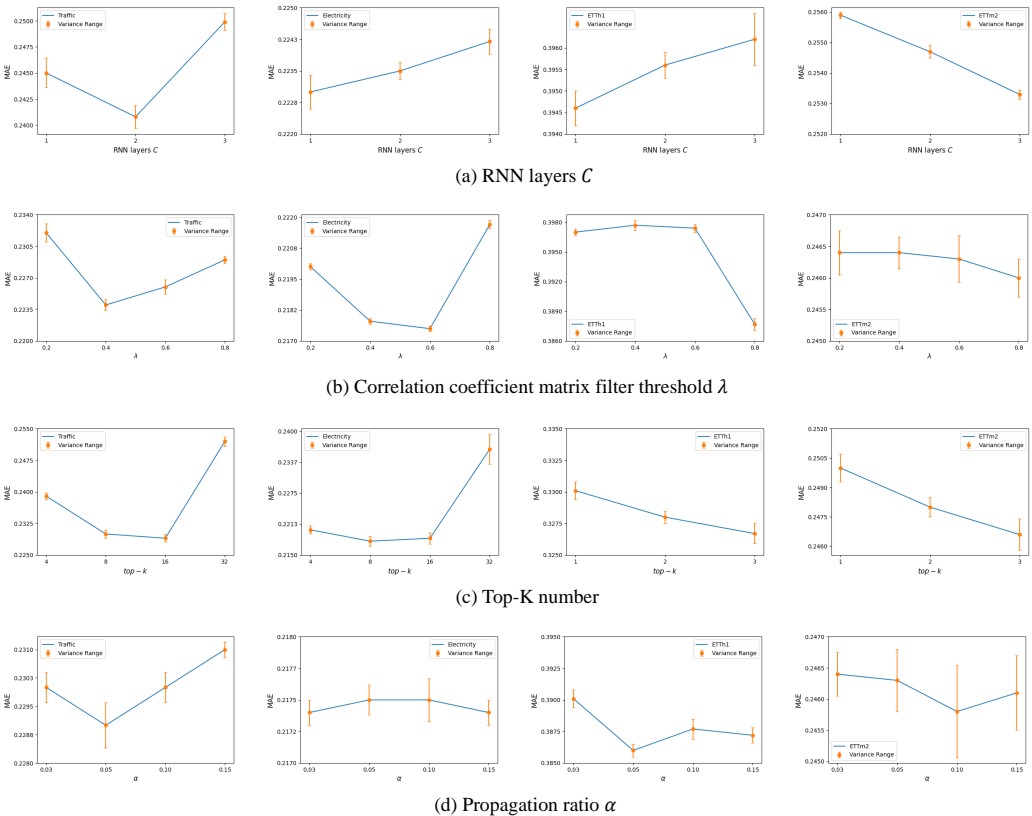

(a) RNN layers $C$

(b) Correlation coefficient matrix filter threshold $\lambda$

(c) Top-K number

(d) Propagation ratio $\alpha$

Figure 4: Hyperparameter sensitivity of (a) RNN layers $C$, (b) coefficient matrix filter threshold $\lambda$, (c) top-k number for selecting edges in the adjacency matrix and (d) propagation ratio $\alpha$ for aggregating information of different hops.

### A.5.3 TOP-K NEIGHBORS

The top-k value is to further filter the sequence pairs that meet the threshold. This allows a sequence to focus on the top-k sequence that is most relevant to it. A small top-k value may result in insufficient sequence interaction information being used, while a large top-k value may make a sequence referencing more relatively weakly correlated neighbors and introduce noise to some extent. To make sure that there are enough sequence pairs to choose from, we fix the value $\lambda$ to 0.4 and try different top-k values for different datasets. For Traffic and Electricity, we select top-k from $\{4, 8, 16, 32\}$. For ETTh1 and ETTm2, we select top-k from $\{1, 2, 3\}$. The results can be seen in Figure 4 (c).

### A.5.4 PROPAGATION RATIO

The propagation ratio $\alpha$ is used to maintain a part of the original embeddings of a patch without information from its multi-hop neighbors. A large value of $\alpha$ constrains the sequence from exploring its neighborhood. Overall, $\alpha$ causes slight difference on the model performance on different datasets, in most cases, the smallest $\alpha$ can't make the model to get the optimal results, therefore it is still necessary to preserve a part of the original embeddings. The results are shown in Figure 4 (d).

### A.6 MODEL EFFICIENCY

We compare GRformer's predictive performance, training speed, and memory footprint to the recognized deep predictive models. The results are recorded using the official model configuration and the same batch size 32. We visualized the model efficiency under datasets Traffic and ETTh1 in Figure 5. Compared to the current most advanced model PatchTST (Nie et al., 2023), GRformer

consumes 91.7% more training time and 99.7% in ETTh1, and 88% more training time and 49.6% more memory in Traffic. This is mainly caused by the use of multi-hop GNN module, because the gradients from multi-hop neighbors need to be storaged and backpropagated with different weights.

Considering the recursive nature of RNN, we also conduct experiments using different RNN layers to see how much computing resource does the RNN position encoding cost. As shown in Table 9, the conditions are different with different layers of RNN. For simple one-layer RNN position encoding (we use 'R-c' to represent experimental settings with c-layer RNN), the model has almost the same training speed and consumes 3.8% more memory than learnable position encoding in ETTh1 on average, and needs 5.9% more training time and 5.3% more memory in Traffic. The gap between RNN-based and learnable position encoding increase as the input sequence length $L$ and the layer number $C$ of RNN become larger. Overall, RNN-based position encoding doesn't result in too much computational resource consumption. This indirectly proves that the main computing resource consumption of the model is concentrated in the GNN module.

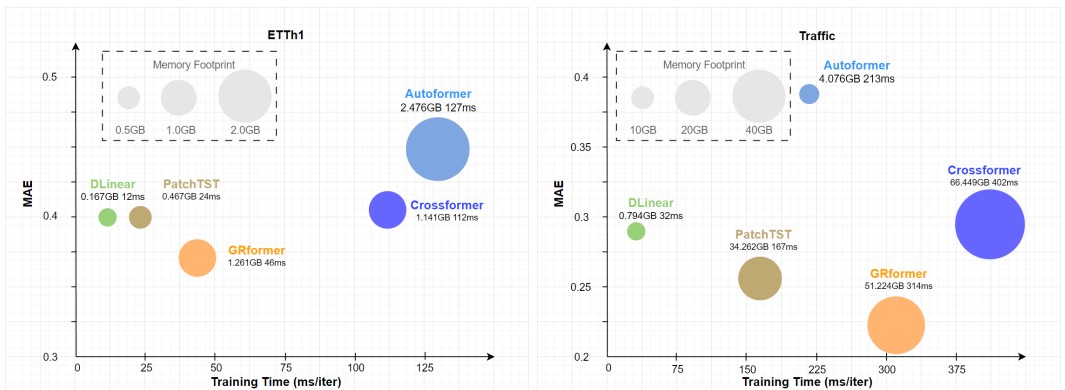

Figure 5: Model efficiency comparison with input length $L = 336$ and $batch\_size = 32$.

Table 9: The training speed and memory consumption of GRformer using only RNN-based position encoding with different input length $L$ and RNN layer depth $C$. dataset-$L$ means the results are recorded in a dataset using the input length of $L$. R-$C$ denotes the model using $C$-layer RNN for generating position encoding. L represents the model using learnable position encoding, which is the same as PatchTST.

| Model | | R-1 | R-2 | R-3 | L |
|---|---|---|---|---|---|
| ETTh1-96 | speed(ms) | 12 | 13 | 14 | 12 |
| | mem(GB) | 0.272 | 0.292 | 0.293 | 0.267 |
| ETTh1-336 | speed(ms) | 24 | 26 | 28 | 24 |
| | mem(GB) | 0.562 | 0.583 | 0.596 | 0.532 |
| ETTh1-720 | speed(ms) | 34 | 37 | 41 | 31 |
| | mem(GB) | 0.850 | 0.944 | 1.181 | 0.818 |
| Traffic-96 | speed(ms) | 83 | 83 | 97 | 74 |
| | mem(GB) | 11.875 | 13.207 | 14.063 | 4.678 |
| Traffic-336 | speed(ms) | 172 | 212 | 239 | 167 |
| | mem(GB) | 40.978 | 54.818 | 50.386 | 38.928 |
| Traffic-720 | speed(ms) | 481 | 493 | 518 | 452 |
| | mem(GB) | 127.242 | 145.046 | 146.172 | 119.672 |

## A.7 VARING LOOK-BACK WINDOW

We set different length of look-back window to test our model's ability of capturing long-term dependencies, and explore whether there is underfitting or overfitting phenomenon. For ILI, we set $L \in \{24, 64, 104, 144\}$, for other datasets, we set $L \in \{96, 192, 336, 720\}$. The results are shown in Table 10.

## A.8 DIVERSE GRAPH CONSTRUCTION METHODS

Besides using Pearson correlation coefficient, there are many more approches for constructing a graph using the raw data, such as DTW (Sakoe & Chiba, 1978) and learnable adjacent matrix. We

Table 10: The prediction performance of GRformer with different historical look-back window. '-' in the table means out of memory.

| $L(\tau)$ | | 96(24) | | 192(64) | | 336(104) | | 720(144) | |
|---|---|---|---|---|---|---|---|---|---|
| Metric | | MSE | MAE | MSE | MAE | MSE | MAE | MSE | MAE |
| Weather | 96 | 0.173 | 0.205 | 0.157 | 0.192 | 0.147 | 0.184 | 0.145 | 0.182 |
| | 192 | 0.219 | 0.247 | 0.202 | 0.234 | 0.192 | 0.232 | 0.189 | 0.227 |
| | 336 | 0.275 | 0.291 | 0.258 | 0.277 | 0.245 | 0.273 | 0.244 | 0.270 |
| | 720 | 0.354 | 0.341 | 0.335 | 0.331 | 0.318 | 0.325 | 0.315 | 0.324 |
| Traffic | 96 | 0.446 | 0.258 | 0.383 | 0.231 | 0.363 | 0.224 | - | - |
| | 192 | 0.454 | 0.261 | 0.401 | 0.237 | 0.385 | 0.232 | - | - |
| | 336 | 0.478 | 0.275 | 0.419 | 0.244 | 0.394 | 0.239 | - | - |
| | 720 | 0.496 | 0.281 | 0.460 | 0.267 | 0.433 | 0.258 | - | - |
| Electricity | 96 | 0.162 | 0.243 | 0.134 | 0.221 | 0.126 | 0.217 | 0.127 | 0.217 |
| | 192 | 0.172 | 0.253 | 0.149 | 0.236 | 0.142 | 0.234 | 0.144 | 0.233 |
| | 336 | 0.187 | 0.269 | 0.164 | 0.252 | 0.155 | 0.248 | 0.160 | 0.249 |
| | 720 | 0.228 | 0.310 | 0.201 | 0.290 | 0.184 | 0.276 | 0.195 | 0.278 |
| ETTh1 | 96 | 0.375 | 0.386 | 0.372 | 0.384 | 0.365 | 0.387 | 0.367 | 0.396 |
| | 192 | 0.424 | 0.417 | 0.415 | 0.413 | 0.406 | 0.412 | 0.403 | 0.421 |
| | 336 | 0.462 | 0.435 | 0.447 | 0.429 | 0.430 | 0.429 | 0.424 | 0.426 |
| | 720 | 0.471 | 0.462 | 0.442 | 0.454 | 0.429 | 0.452 | 0.427 | 0.454 |
| ETTh2 | 96 | 0.280 | 0.330 | 0.278 | 0.331 | 0.274 | 0.332 | 0.273 | 0.334 |
| | 192 | 0.357 | 0.379 | 0.345 | 0.376 | 0.337 | 0.373 | 0.339 | 0.375 |
| | 336 | 0.399 | 0.412 | 0.374 | 0.399 | 0.355 | 0.390 | 0.366 | 0.398 |
| | 720 | 0.409 | 0.430 | 0.392 | 0.419 | 0.382 | 0.417 | 0.404 | 0.435 |
| ETTm1 | 96 | 0.311 | 0.339 | 0.294 | 0.329 | 0.281 | 0.326 | 0.286 | 0.331 |
| | 192 | 0.362 | 0.367 | 0.333 | 0.355 | 0.325 | 0.356 | 0.329 | 0.359 |
| | 336 | 0.293 | 0.289 | 0.363 | 0.378 | 0.357 | 0.377 | 0.362 | 0.380 |
| | 720 | 0.456 | 0.427 | 0.425 | 0.417 | 0.417 | 0.413 | 0.409 | 0.410 |
| ETTm2 | 96 | 0.173 | 0.252 | 0.166 | 0.274 | 0.161 | 0.246 | 0.161 | 0.248 |
| | 192 | 0.239 | 0.296 | 0.225 | 0.288 | 0.213 | 0.284 | 0.223 | 0.292 |
| | 336 | 0.301 | 0.336 | 0.277 | 0.323 | 0.266 | 0.319 | 0.269 | 0.322 |
| | 720 | 0.403 | 0.393 | 0.368 | 0.379 | 0.351 | 0.372 | 0.347 | 0.374 |
| ILI | 24 | 0.331 | 1.102 | 1.459 | 0.743 | 1.281 | 0.731 | 1.446 | 0.775 |
| | 36 | 2.067 | 0.884 | 1.557 | 0.767 | 1.104 | 0.671 | 1.363 | 0.747 |
| | 48 | 1.999 | 0.884 | 1.602 | 0.802 | 1.255 | 0.718 | 1.350 | 0.769 |
| | 60 | 2.122 | 0.925 | 0.763 | 0.851 | 1.314 | 0.737 | 1.486 | 0.811 |

conduct exporiments to see the difference of varying graph constructing methods. For DTW algorithms, we calculate the distance $Distance_{(i,j)}$ between the $i$-th and $j$-th univariant time series, and use $\frac{1}{1+Distance_{(i,j)}}$ as the correlation coefficient between them. For learnable adjacency matrix, we randomly initialize an adjacency matrix $\mathbf{A}_\theta \in \mathbb{R}^{C \times C}$ and parameterize it. We fix other hyperparameters to fairly compare these three methods.

All the results can be seen in Table 11. Although DTW is more complex than Pearson correlated coefficient algorithm, it does not have obvious significant performance improvement. Learnable adjacency matrix performs better in some cases, and offer better flexibility, however, it costs too much computing resources, and has a relatively slow training speed. Overall, the use of Pearson correlation coefficient method is general and effective. In more complex application scenarios, other graph constructing algorithms may generate better graph structures. Our graph constructing module is independent of the training process, which makes the substitution of algorithms very convenient and flexible.

### A.9 ROBUSTNESS ANALYSIS

All results in the main text and appendix above are obtained using the fixed random seed 2023. We train GRformer with three other random seeds to evaluate the robustness of our results, as shown in Table 12. It can be seen that the variances are small, indicating the robustness against choice of random seeds of our model.

### A.10 VISUALIZATION

We visualize the long-term forecasting results of GRformer, PatchTST, DLinear, and Autoformer in Figure 6. We predict 192 steps on ETTm2 and 60 steps on ILI. GRformer provides stable prediction results which is the closest to the ground truth.

Table 11: The prediction performance of GRformer with different graph constructing methods: (a) DTW algorithm, (b) learnable adjacency matrix and (c) Pearson correlated coefficient algorithm. '-' in the table means out of memory.

| Models | | DTW | | Learnable | | Pearson | |
|---|---|---|---|---|---|---|---|
| Metric | | MSE | MAE | MSE | MAE | MSE | MAE |
| Weather | 96 | 0.149 | 0.189 | 0.142 | 0.184 | 0.147 | 0.184 |
| | 192 | 0.195 | 0.232 | 0.186 | 0.228 | 0.192 | 0.232 |
| | 336 | 0.246 | 0.273 | 0.237 | 0.253 | 0.245 | 0.273 |
| | 720 | 0.320 | 0.327 | 0.284 | 0.288 | 0.318 | 0.325 |
| Traffic | 96 | 0.370 | 0.228 | - | - | 0.363 | 0.224 |
| | 192 | 0.389 | 0.237 | - | - | 0.385 | 0.232 |
| | 336 | 0.400 | 0.246 | - | - | 0.394 | 0.239 |
| | 720 | 0.446 | 0.267 | - | - | 0.433 | 0.258 |
| Electricity | 96 | 0.128 | 0.219 | 0.124 | 0.213 | 0.126 | 0.217 |
| | 192 | 0.145 | 0.235 | 0.144 | 0.230 | 0.142 | 0.234 |
| | 336 | 0.161 | 0.253 | 0.160 | 0.252 | 0.155 | 0.248 |
| | 720 | 0.186 | 0.277 | 0.190 | 0.284 | 0.184 | 0.276 |
| ETTh1 | 96 | 0.373 | 0.392 | 0.360 | 0.381 | 0.365 | 0.387 |
| | 192 | 0.410 | 0.417 | 0.406 | 0.412 | 0.406 | 0.412 |
| | 336 | 0.432 | 0.431 | 0.433 | 0.426 | 0.430 | 0.429 |
| | 720 | 0.444 | 0.458 | 0.432 | 0.450 | 0.429 | 0.452 |
| ETTh2 | 96 | 0.275 | 0.334 | 0.269 | 0.324 | 0.274 | 0.332 |
| | 192 | 0.337 | 0.374 | 0.337 | 0.373 | 0.337 | 0.373 |
| | 336 | 0.356 | 0.389 | 0.353 | 0.391 | 0.355 | 0.390 |
| | 720 | 0.384 | 0.419 | 0.388 | 0.424 | 0.382 | 0.417 |
| ETTm1 | 96 | 0.285 | 0.329 | 0.289 | 0.326 | 0.284 | 0.326 |
| | 192 | 0.332 | 0.361 | 0.321 | 0.358 | 0.325 | 0.356 |
| | 336 | 0.361 | 0.379 | 0.358 | 0.391 | 0.357 | 0.377 |
| | 720 | 0.426 | 0.415 | 0.419 | 0.414 | 0.417 | 0.413 |
| ETTm2 | 96 | 0.161 | 0.246 | 0.158 | 0.240 | 0.161 | 0.246 |
| | 192 | 0.218 | 0.285 | 0.210 | 0.280 | 0.213 | 0.284 |
| | 336 | 0.272 | 0.328 | 0.265 | 0.318 | 0.266 | 0.319 |
| | 720 | 0.354 | 0.372 | 0.349 | 0.370 | 0.351 | 0.372 |
| ILI | 24 | 1.414 | 0.736 | 1.324 | 0.720 | 1.281 | 0.731 |
| | 36 | 1.362 | 0.699 | 1.514 | 0.757 | 1.104 | 0.671 |
| | 48 | 1.302 | 0.807 | 1.542 | 0.864 | 1.255 | 0.718 |
| | 60 | 1.621 | 0.825 | 1.283 | 0.701 | 1.314 | 0.737 |

Table 12: Multivariate long-term forecasting results with different random seeds in different datasets.

| Model | | GRformer | |
|---|---|---|---|
| Metric | | MSE | MAE |
| Weather | 96 | 0.1475±0.0002 | 0.1843±0.0004 |
| | 192 | 0.1926±0.0006 | 0.231±0.0005 |
| | 336 | 0.2463±0.0014 | 0.2246±0.0011 |
| | 720 | 0.3188±0.0008 | 0.3259±0.0005 |
| Traffic | 96 | 0.3637±0.0011 | 0.2243±0.0004 |
| | 192 | 0.3853±0.0008 | 0.3256±0.0007 |
| | 336 | 0.3957±0.0014 | 0.2401±0.0008 |
| | 720 | 0.4337±0.0006 | 0.2622±0.0008 |
| Electricity | 96 | 0.1262±0.0002 | 0.2173±0.0002 |
| | 192 | 0.1422±0.0005 | 0.2347±0.0004 |
| | 336 | 0.1554±0.0003 | 0.2488±0.0006 |
| | 720 | 0.1864±0.0022 | 0.2773±0.0012 |
| ETTh1 | 96 | 0.3655±0.0010 | 0.3868±0.0005 |
| | 192 | 0.4048±0.0006 | 0.4113±0.0008 |
| | 336 | 0.4312±0.0012 | 0.4305±0.0010 |
| | 720 | 0.4369±0.0073 | 0.4585±0.0056 |
| ETTh2 | 96 | 0.2751±0.0007 | 0.3330±0.0005 |
| | 192 | 0.3378±0.0008 | 0.3743±0.0012 |
| | 336 | 0.3562±0.0016 | 0.3907±0.0004 |
| | 720 | 0.3846±0.0025 | 0.4229±0.0013 |
| ETTm1 | 96 | 0.2853±0.0008 | 0.3265±0.0003 |
| | 192 | 0.3254±0.0006 | 0.3571±0.0012 |
| | 336 | 0.3601±0.0029 | 0.3786±0.0012 |
| | 720 | 0.4223±0.0052 | 0.4141±0.0014 |
| ETTm2 | 96 | 0.1608±0.0002 | 0.2460±0.0001 |
| | 192 | 0.2142±0.0010 | 0.2842±0.0003 |
| | 336 | 0.2688±0.0022 | 0.3207±0.0014 |
| | 720 | 0.3515±0.0008 | 0.3728±0.0005 |
| ILI | 24 | 1.3075±0.0333 | 0.7337±0.0026 |
| | 36 | 1.3392±0.1844 | 0.7293±0.0521 |
| | 48 | 1.3731±0.1470 | 0.7479±0.0314 |
| | 60 | 1.4249±0.0842 | 0.7742±0.0280 |

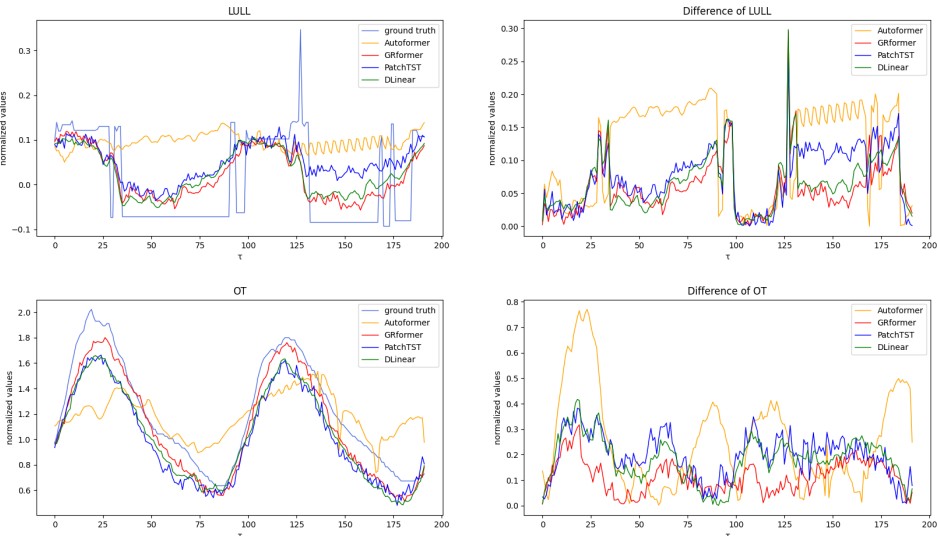

(a) The prediction results on ETTm2. The chosen variables are 'LULL' and 'OT'.

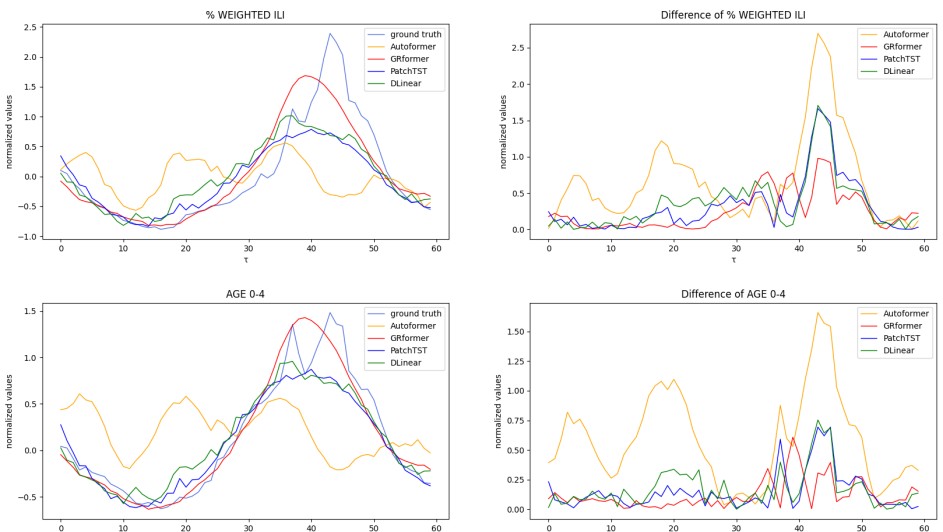

(b) The prediction results on ILI. The chosen variables are '%WEIGHTED ILI' and 'AGE 0-4'.

Figure 6: The visulization of (a) 192-step prediction results on ETTm2 where $L = 336$, and (b) 60-step prediction results on ILI where $L = 104$. The variables chosen are 'Low UseLess Load (LULL)' and 'Oil Temperature (OT)' in ETTm2 and '% WEIGHTED ILI' and 'AGE 0-4' in ILI. GRformer can offer the closest prediction results to the ground truth (in light blue).

