# OpenReview forum: "Connecting the Patches: Multivariate Long-term Forecasting using Graph and Recurrent Neural Network"
_ICLR.cc/2024/Conference — Submitted to ICLR 2024_

### Official Review · Reviewer_qMLP · 2023-10-22

**Soundness:** 3 good
**Presentation:** 3 good
**Contribution:** 3 good
**Rating:** 6
**Confidence:** 2

**Summary:**

This paper delves into the challenges presented by multivariate long-term time series forecasting (MLTSF), specifically the difficulty of capturing cross-channel dependencies and temporal order information using current Transformer-based models. Despite the achievements of Transformer models in various fields, their application in MLTSF reveals certain inadequacies. Models like Informer, Autoformer, and FEDformer, while advanced, still face challenges in understanding intricate channel relationships in multivariate time series.

To address these issues, the authors propose the GRformer model. This innovative solution combines the strengths of Graph Neural Networks (GNN) and position encoding derived from Recurrent Neural Networks (RNN). The inclusion of a mix-hop propagation layer within a feedforward neural network promotes efficient interaction between different time series data points. Additionally, by leveraging a multi-layer RNN, the model recursively generates positional embeddings, emphasizing the importance of sequence order.

The paper's empirical tests, conducted on eight real-world datasets, demonstrate the GRformer's superior predictive accuracy in MLTSF tasks, underlining its potential as a novel solution in the field of time series forecasting.

**Strengths:**

**Strengths**:

1. **Originality**:
   - The GRformer presents a unique fusion of GNN and RNN-based position encoding within a Transformer framework, addressing gaps in MLTSF.
   - The incorporation of the Pearson correlation coefficient for graph structure is a notable innovation.

2. **Quality**:
   - Rigorous empirical validation is conducted on eight real-world datasets, ensuring robustness.
   - The model's design is comprehensive, with the mix-hop propagation layer and RNN-based position encoding as highlights.

3. **Clarity**:
   - The paper delineates complex concepts coherently, facilitating reader understanding.
   - Distinctive features and advantages of GRformer over existing models are clearly articulated.

4. **Significance**:
   - The GRformer's advancements in capturing cross-channel dependencies have potential broad impacts in time series forecasting.
   - The paper paves the way for future research by highlighting existing challenges and areas of improvement.

In essence, the paper excels in its innovative methodology, thorough validation, lucid presentation, and relevance in the field.

**Weaknesses:**

1. **Mathematical Notation Consistency**:
   - The authors' use of mathematical notation appears inconsistent. For instance, function names should ideally be presented in regular typeface rather than italic. Proper notation ensures clarity and avoids potential confusion.

2. **Graph Construction Using Pearson Coefficient**:
   - While the authors opted for the Pearson correlation coefficient for graph construction, which subsequently serves as the foundational structure for the GNN, one might question the exclusion of making GNN parameters learnable. This adaptability could potentially offer more flexibility to the model.

3. **Assumption of Homoscedasticity**:
   - The Pearson coefficient assumes homoscedasticity in the data. It's unclear if the authors verified this assumption across their datasets. Such checks are crucial to ensure the validity of the chosen coefficient.

4. **Alternative Correlation Metrics**:
   - The paper doesn't seem to explore or discuss other potentially beneficial correlation coefficients like Time-Lagged Cross-Correlation (TLCC) or Dynamic Time Warping (DTW). An exploration or justification of the chosen metric over others could have added depth to their methodology.

**Questions:**

**Hyperparameter Selection in Graph Construction**:
   - The methodology introduced by the authors involves several hyperparameters, which seemingly have a significant impact on the model's outcomes. Specifically, when constructing the graph structure:
     - How was the threshold value of 0.8 determined?
     - Regarding the 'topk' selection, how was the value of \( k \) chosen, and does it correlate with the number of variables?

 **Mix-hop Propagation Parameter**:
   - How was the value for the EMA parameter \( \alpha \) in the mix-hop propagation process determined?

---

> ### Author Response · Authors · 2023-11-21
> **Response to Reviewer qMLP (Q5, Q6)**
>
> > Q5: Hyperparameter Selection in Graph Construction:
> > - The methodology introduced by the authors involves several hyperparameters, which seemingly have a significant impact on the model's outcomes. Specifically, when constructing the graph structure:
> >     - How was the threshold value of 0.8 determined?
> >     - Regarding the 'topk' selection, how was the value of ( k ) chosen, and does it correlate with the number of variables?
>
> A5: Thank you for your question. The threshold (we use $\lambda$ in the following text) 0.8 and top-k value are chosen according to experience in the earlier version. Based on your comment, we set these two parameters as hyperparameters and conduct parameter sensitivity analysis on different datasets. We find that the value settings in the early version do not guarantee optimal results, in the revised version, we update the parameter sensitivity analysis results to **appendix A.5.2** and **A.5.3**. Overall, using different thresholds and topk value under different datasets can achieve better results, e.g. for traffic, the best $\lambda$ is 0.6 and the best top-k is 16.
> Top-k value does not have direct correlation with the number of variables, however, based on the multi-hop depth $gdep$, we tend to choose top-k values that meet the following condition: (top-k)$^{gdep}\le M$, where $M$ is the number of variables.
>
> > Q6: Mix-hop Propagation Parameter:
> > - How was the value for the EMA parameter ( $\alpha$ ) in the mix-hop propagation process determined?
>
> A6: Thank you for your question. We also conduct parameter sensitivity analysis for the value of $\alpha$ in the revised version, and choose the optimal results. We have updated these results to **appendix (A.5.4)**.

---

> ### Author Response · Authors · 2023-11-21
> **Response to Reviewer qMLP (Q1, Q2, Q3 and Q4)**
>
> > Q1: Mathematical Notation Consistency:
> > - The authors' use of mathematical notation appears inconsistent. For instance, function names should ideally be presented in regular typeface rather than italic. Proper notation ensures clarity and avoids potential confusion.
>
> A1: We appreciate your comment for helping to optimize our paper. We check the main text and appendix of our paper and make corrections to address this issue. We modify the font of the function and variable letters, and correct the italicized font of the function to normal, such as covariance function, argtopk, softmax function.
>
> > Q2: Graph Construction Using Pearson Coefficient:
> > - While the authors opted for the Pearson correlation coefficient for graph construction, which subsequently serves as the foundational structure for the GNN, one might question the exclusion of making GNN parameters learnable. This adaptability could potentially offer more flexibility to the model.
>
> A2: We appreciate your suggestion. Parameterizing the graph structure to achieve greater flexibility is a worthwhile try. We parameterize the graph structure to see the performance changes of GRformer, and the results show that parameterizing the graph structure achieves better performance on some of the datasets(e.g. Weather), but the effect is not significant on most datasets, and even inferior to simple correlation coefficient algorithms. We have updated the results to **appendix (A.8)**.
>
> >  Q3: Assumption of Homoscedasticity:
> > - The Pearson coefficient assumes homoscedasticity in the data. It's unclear if the authors verified this assumption across their datasets. Such checks are crucial to ensure the validity of the chosen coefficient.
>
> A3: Thank you for your suggestion. The Pearson correlation coefficient is calculated based on the standardized data. We use the Bartlett function in the Python 'scipy' library to verify the homoscedasticity of the data we use, and obtaine test statistics and test p on 8 datasets, respectively. The results indicate that the statistical values of these data are very small, and p_value is close to 1.0, which makes us confident that the data we use satisfies homoscedasticity. We provide additional clarification on this issue in **appendix (A.5)**.
>
> > Q4: Alternative Correlation Metrics:
> > - The paper doesn't seem to explore or discuss other potentially beneficial correlation coefficients like Time-Lagged Cross-Correlation (TLCC) or Dynamic Time Warping (DTW). An exploration or justification of the chosen metric over others could have added depth to their methodology.
>
> A4: Thank you for your suggestion! We agree with you with the insight that more attempts on the Graph construction module is beneficial. Based on your suggestion, we try some other graph structure construction algorithms, such as the DTW algorithm, and update the results in **appendix (A.8)**. From the results, although the DTW algorithm has a more complex and refined calculation process compared to the Pearson coefficient, there is not significant performance improvement. The default graph construction method in our proposed GRformer is still relatively simple and widely effective. TLCC generates a correlation matrix similar to Pearson coefficient method, so we do not conduct experiments with this setting.
> We also clearly recognize that in the face of more complex scenarios, using different algorithms may also result in better performance. Our graph construction module is independent of the training process, making it easy to be replaced with other graph construction algorithms to generate reasonable graph structures.

---

### Official Review · Reviewer_Evwp · 2023-10-23

**Soundness:** 2 fair
**Presentation:** 3 good
**Contribution:** 2 fair
**Rating:** 3
**Confidence:** 4

**Summary:**

This paper enhances Transformer with GNN and position embedding generated by RNN for multivariate time series forecasting. The proposed GRformer constructs graph by pearson correlation and uses a mix-hop propagation GNN layer to capture cross-channel dependency. For temporal dependency, it uses an RNN to recursively generate positional embeddings. Experiments on eight real-world datasets show that the proposed GRformer is on compare with SOTA model, PatchTST.

**Strengths:**

- This paper is well-written and easy to follow.
- Using pearson correlation for graph constructing is reasonable and efficient.

**Weaknesses:**

My main concern is that the novelty is limited:

- For RNN-based position embedding:
  1. The idea of enhance Transformer with RNN is not new[1].
  2. RNN operates recursively and cannot be parallelized, which offsets the efficiency advantages of Transformers that can be highly parallelized.
  3. Ablation study in Table 3 shows that the improvement of RNN against previous learnable position embedding is not significant.
- For Mix-hop propagation:
    1. The mix-hop propagation layer is **exactly the same** as that in [2] and there is no explicit reference to it in Section 3.2.3.
    2. Besides the graph construction via Pearson correlation, this is a direct combination of PatchTST and "Connecting the dots".

[1] Qin, Yao, et al. "A dual-stage attention-based recurrent neural network for time series prediction." arXiv preprint arXiv:1704.02971 (2017).

[2] Wu, Zonghan, et al. "Connecting the dots: Multivariate time series forecasting with graph neural networks." Proceedings of the 26th ACM SIGKDD international conference on knowledge discovery & data mining. 2020.

**Questions:**

- What is the authors' primary objective in visualizing the weights of the MLP in Figure 1(b), given that it only reflects the correlation among hidden states?
- Could you provide a comparison of the computational efficiency between your RNN-based position embedding and a learnable position embedding, particularly in relation to varying sequence lengths?
- How were the hyperparameters (0.8 and $k$) in Equations (2) and (3) chosen, and what impact do these specific values have on the model's performance and behavior?

---

> ### Author Response · Authors · 2023-11-21
> **Response to Reviewer Evwp (Q3, Q4 and Q5)**
>
> > Q3: What is the authors' primary objective in visualizing the weights of the MLP in Figure 1(b), given that it only reflects the correlation among hidden states?
>
> A3: Thanks for asking the question. We modify the corresponding figures and explanations of **Figure 1(b)** to make it clear. Values of each channel of a multi-variant time series will be passed through an embedding mapping projector, a series of feedforward networks of Encoders and Decoders, and a final mapping projector. These neural networks are usually some MLP layers, such as Linear layer and Conv1d layer. The weight matrix plays an important part in promoting interactions across different channels. We save the weight matrix parameters (we choose parameters from Informer) and calculate their matrix products, and get an equivalent correlation matrix. Compared with the Pearson correlation coefficient matrix (or the correlation matrix obtained by any other algorithms), the result does not effectively promote the interaction between strong correlated sequences and prevent weak correlated sequences from interacting, either.
>
> > Q4: Could you provide a comparison of the computational efficiency between your RNN-based position embedding and a learnable position embedding, particularly in relation to varying sequence lengths?
>
> A4: Thank you for your suggestion. Calculating the efficiency of the module does rigorously evaluate our design better. Under your suggestion, we visualized the memory usage and training speed of GRformer and other models under the same batch size (we choose 32). We remove the graph convolutional module and use RNN position encoding only to illustrate the impact of introducing RNN on model efficiency. The results indicate that the RNN structure does prolong the training time, but it is not much and is related to the length of the input sequence $L$ and depth of RNN structure $C$. The specific results are shown in **appendix (A.6)**, **Table 9** in the revised version. In relatively small datasets such as ETTh1, the use of 1-layer RNN position encoding increase the memory usage by 3.8% compared to patchTST, and the training time is almost not increased; In relatively large datasets such as Traffic, the memory usage increases 12.6% and training time increases 5.9%. With a larger value of RNN depth $C$ and input length $L$, the efficiency gap will increase, the growth rate is within an acceptable range.
>
> > Q5: How were the hyperparameters (0.8 and $k$) in Equations (2) and (3) chosen, and what impact do these specific values have on the model's performance and behavior?
>
> A5: We appreciate you for this detailed question. The choice of 0.8 and top-k value in the previous version of our paper are based on experience. Based on your comment, in order to provide a more rigorous explaination of our hyperparameter choosing, we conduct hyperparameter sensitivity analysis on the filtering threshold (we use $\lambda$ to represent it) and top-k values of the Pearson coefficient correlation matrix, and update the results to **appendix A.5.2, A.5.3**, and **A.5.4**. A smaller threshold and larger top-k value will allow for more interactions between channels. Directly choosing 0.8 based on experience does have some problems and changing this threshold to smaller values can achieve better results on different datasets(e.g. GRformer gets the optimal results using $\lambda=0.6$ and top-k $=16$ in Traffic). We choose the best results and update **Table 2** in **section 4.1**.

---

> ### Author Response · Authors · 2023-11-21
> **Response to Reviewer Evwp (Q1 and Q2)**
>
> > Q1: For RNN-based position embedding:
> > - The idea of enhance Transformer with RNN is not new[1].
> > - RNN operates recursively and cannot be parallelized, which offsets the efficiency advantages of Transformers that can be highly parallelized.
> > - Ablation study in Table 3 shows that the improvement of RNN against previous learnable position embedding is not significant.
>
> A1: We appreciate for your comment.
> - There are indeed some previous studies using RNN to enhance Transformer, and we have noted the work you mentioned. But the way we use RNN is different. In paper [1], Attention machenism is used to generate an enhanced input sequence to be fed in an RNN structure, with its attention score generated with the help of the previous RNN hidden states. The encoder is essentially an RNN that encodes the input sequences into a feature representation, which makes the model still suffer from gradient explosion and vanishing when facing long-term sequence.  In GRformer, we do not directly pass input data through the RNN layer, we only use RNN to explicitly introduce temporal order information in position encodings that are injected into sequence tokens, driven by the features. The encoder of GRformer is essentially a self-attention layer. To our knowledge, there are currently no similar position encoding methods like this to solve MLTSF tasks, and our work also explores the effectiveness of this approach.
> - As for the model efficiency, we conduct experiments to visualize the training speed and memory consumption of using RNN-based position encoding. In small dataset like ETTh1, a one-layer RNN-based position encoding module costs almost the same training time and 3.8% more memory compared with learnable position encoding (i.e. PatchTST), while the gap comes to 5.9% more training time and 5.3% more memory in Traffic. The results are added to **appendix (A.6)**.
> - We evaluate the effectiveness of using RNN-based position encoding with varying depth on different datasets and add the results in **appendix (A.5.1)*, overall, RNN-based position encoding can achieve 2.51% and 1.83% reduction in MSE and MAE. We also put results of ablation studies on all datasets in **section 4.2**.
>
>
> > Q2: For Mix-hop propagation:
> > - The mix-hop propagation layer is exactly the same as that in [2] and there is no explicit reference to it in Section 3.2.3.
> > - Besides the graph construction via Pearson correlation, this is a direct combination of PatchTST and "Connecting the dots".
>
> A2: Thank you for your comment.
> - Yes, we do draw inspiration from MTGNN[1] for using multi-hop propagation layers. We explicit reference to it in **section 3.2.3** in the revised version. However, our multi-hop propagation layer has some differences in implementation details compared to MTGNN. In MTGNN, embeddings from each hop are aggregated through a Conv2d layer with a convolutional kernel size of (1,1), which means that the aggregated embeddings of each channel in each hop are all viewed as separate individuals (i.e. they will be assigned with different weights separately). We choose to use a learnable matrix to aggregate the embeddings of different hop neighbors, so that embeddings of each channel from the same hop neighbors will be assigned with the same weights, this makes it easier for the model to identify the importance of neighborhoods’ information of different hops.
> - One of our motivation is that, the current channel-mixing and channel-independent models can’t effectively capture the cross-channel dependencies between time series, while graph neural network provides a chance to make better solutions. We explored the effectiveness of graph neural networks in MLTSF tasks, and from the results in **Table 3**, it improves the MSE and MAE by 0.65% and 3.63% respectively.
>
>
> [1] Qin, Yao, et al. "A dual-stage attention-based recurrent neural network for time series prediction." arXiv preprint arXiv:1704.02971 (2017).
>
> [2] Wu, Zonghan, et al. "Connecting the dots: Multivariate time series forecasting with graph neural networks." Proceedings of the 26th ACM SIGKDD international conference on knowledge discovery & data mining. 2020.

---

> > ### Comment · Reviewer_Evwp · 2023-11-22
> > **Response to Authors' Rebuttal**
> >
> > Thanks for your feedback, which addresses my concern about hyper-parameters of graph construction. However, my other concerns have not been addressed yet:
> > 1) The modified Figure 1(b) reflects nothing about channel dependency: there are many non-linear operations in the network, such as activations and self-attentions, so it makes no sense to multiply all weight matrices together.
> > 2) RNN-based position encoding goes against the original intention of position encoding, which is to introduce positional information to the transformer in a non-recursive way.
> > 3) Despite the aggregation matrix being slightly different, there is no big difference between the mix-hop propagation in this paper and  "Connecting the dots", thus it should not be listed as a contribution.
> >
> > Based on the above reasons, I'll maintain my score of 3.

---

> > > ### Author Response · Authors · 2023-11-22
> > > **Response to Reviewer Evwp (Response to the Rebuttal)**
> > >
> > > Thank you for your comment. We carefully consider the issues you raised and will address them from the following three aspects:
> > > 1. **About the meanings of Figure 1(b).** The product of the weight matrix in Figure 1(b) is equivalently applied to each variable at each timestamp, so we believe that the weight matrix product could reflect channel dependencies to some extent. Although the self-attention layer and the non-linear projectors change the embeddings of each timestamp, they don't change the fact that the embedding contains the multi-channel information. The embedding of each timestamp contains features of different channels, the self-attention layer makes embeddings of different timestamps interact with each other, after which the embeddings still carry multi-channel information. The non-linear activation function (such as ReLU) may introduce non-linear factors, but the multi-channel information is still retained. These modules indeed cause various changes to the embeddings, but there is no external information been introduced, thus these embeddings still represent or contain the information from different variables. **In this case, the MLP layer of the feedforward network maps the embeddings further, so our understanding of the feedforward network is that while it extracts richer semantic information, it also implicitly promotes the interaction between different channels, which can only be determined by the weight matrix** (i.e. whether the elements in one embedding should interact or not). The product result of the weight matrix may not reflect this fact absolutely accurately, but it does illustrate this issue to some extent.
> > > 2. **About RNN-based module goes against a non-recursive way.** We agree with you that RNN does not work in a parallel way, however, the approach of generating position encoding is not strictly required to be parallel, and there have been some previous works generating position encoding with a recursive way and achieved better results in NLP tasks. R-Transformers[3] uses RNN structures locally to generate  position encodings in a data-driven manner, and FLOATER [4], which does not use RNN structures, attempts to combine with system dynamics theory to make position encodings explicitly carrying temporal order information. These works inspired us to use RNN structures for position encoding. However, as we are dealing with long-term predicting tasks, the above model designs may introduce excessive computational complexity. Therefore, we design and use a multi-layer RNN structure. We analyze the efficiency of RNN module in **appendix (A.5), Figure 5**. We find that a single-layer RNN actually has almost the same computational complexity as a learnable matrix of the same size (i.e., learnable position encoding) and increasing the number of RNN layers does not cause a significant decrease in model efficiency.
> > > 3. **About the mix-hop propagation layer.** We appreciate you for pointing out this issue, and based on your comment, we modify the statement of our contribution in the revised version to make it more clear. For capturing cross-channel dependencies, existing attempts mainly use the Attention mechanism, such as Crossformer[5], but we think that GNN may achieve better results. From the perspective of the GNN structure, we indeed do not make significant alterations. However, the use of GNN involves not only the design of the message propagation layer, but also the construction of the graph structure. In fact, we tried various graph constructing algorithms, such as Pearson coefficient correlation, DTW and learnable adjacency matrix, we add the results in **Table 11** in **appendix (A.8)**. We mainly explore the effect of combining Transformer with GNN and propose this model. In terms of the results, this approach is effective. We add the statement of our graph construction module in the contribution of the revised version.
> > >
> > > [3] Zhiwei Wang, Yao Ma, Zitao Liu, and Jiliang Tang. R-transformer: Recurrent neural network enhanced transformer. arXiv preprint arXiv:1907.05572, 2019.
> > >
> > > [4] Liu, Xuanqing, et al. "Learning to encode position for transformer with continuous dynamical model." International conference on machine learning. PMLR, 2020.
> > >
> > > [5] Zhang, Yunhao, and Junchi Yan. "Crossformer: Transformer utilizing cross-dimension dependency for multivariate time series forecasting." The Eleventh International Conference on Learning Representations. 2022.

---

### Official Review · Reviewer_rJkP · 2023-10-31

**Soundness:** 1 poor
**Presentation:** 2 fair
**Contribution:** 1 poor
**Rating:** 3
**Confidence:** 4

**Summary:**

This paper proposes GRformer, a new neural architecture for multivariate long-term time series forecasting (MLTSF). The authors propose a hybrid architecture that consists of a Transformer-based graph neural network to model cross-channel dependencies and a recurrent neural network to model temporal dependencies. The proposed model shows promising performance on eight benchmarks. However, the motivation and reasoning behind the criticism of the Transformer-based approach are difficult to understand. Some of the claims are made without proper evidence, or by simply citing previous work, without providing any further detailed study or analysis. Additionally, the performance improvements on the benchmarks seem to outperform the baselines. However, I believe the claim of achieving a performance improvement with a 5.7% decrease in MSE and 6.1% decrease in MAE is misleading. These numbers are calculated by averaging MSE and MAE without considering the scales between different benchmarks and metrics. ILI has much higher mean squared errors (MSEs) and mean absolute errors (MAEs) than other benchmarks. This means that if you compute the average score in this way, the average score can be dominated by the relative improvement in this specific dataset. The tone reporting the improvement suggests that the model showed around a 6% decrease in errors on all benchmarks, but the average relative improvement for each benchmark at different metrics is actually 2.55% for MSE and 4.96% for MAE.

**Strengths:**

The model achieves improvements over 7 different benchmarks using 4 metrics for each benchmark dataset. The experiments are done extensively with ablation on different positional encoding strategies. This however raises a question on why the RNN is needed (Table 3. R: the first column vs L: the second column show a very minor difference).

**Weaknesses:**

I am not sure what I am seeing in Figure 1(b), and I don’t understand how to interpret the authors' claim that cross-channel interaction is chaotic based on simply visualizing the weight matrices of the Transformer's dense layer (internal MLP).

I am not sure I understand the authors' point about positional encoding not being able to represent temporal orders well. RNNs have their own problems, such as vanishing gradients when modeling long-term temporal dependencies. Are you suggesting that RNNs outperform Transformers in multivariate long-term time series forecasting (MLTSF)?
-> Are the ablation results in Table 3 the experiments to back this claim? If that's the case, the performance difference between an RNN-based positional encoding (?) vs a learned positional embedding is almost 0.

What exactly is the RNN-based position encoding method? In the caption for Figure 2, it says "The multi-layer RNN injects temporal order information." However, RNNs are not just injecting temporal order information as some sort of advanced positional encoding method; they can actually learn temporal dependencies. I am not sure if you are distinguishing between positional encoding and learning temporal representation.

Figure 2 (b) is hard to understand, at least explain the operator signs in the caption, arrows are not clear.

**Questions:**

What is the main evidence that Transformer-based models are ineffective at capturing cross-channel dependencies and temporal orders? If Transformers were bad at capturing temporal orders, they would not have become as popular as they are today. I am curious why the authors make such claims, as I do not see any plausible supporting evidence in the manuscript.

The authors mentioned that they used multi-layered RNNs, however in the appendix, it's said 1-layer RNN was used. Can you clarify the details of the RNN architecture?

“To properly capture temporal dependencies, we consider using a multilayer RNN to encode the positions in the time series.” Why deep RNNs can properly capture temporal dependencies while Transformers can’t?

---

> ### Author Response · Authors · 2023-11-21
> **Response to Reviewer rJkP(Q5,Q6 and Q7)**
>
> > Q5: What is the main evidence that Transformer-based models are ineffective at capturing cross-channel dependencies and temporal orders? If Transformers were bad at capturing temporal orders, they would not have become as popular as they are today. I am curious why the authors make such claims, as I do not see any plausible supporting evidence in the manuscript.
>
> A5: We appreciate you for this thoughtful question, we agree with you that Transformer is popular now and good at capturing temporal dependencies. We dive deep into the characteristics of time series data and believe that Transformer-based models can be further improved in extracting cross-channel and temporal dependencies.
> 1. **Cross-channel dependencies.** Values of each channel of a multi-variant time series data will be passed through an embedding mapping projector, a series of feedforward networks of Encoders and Decoders, and a final mapping projector. These modules are usually implemented with MLP layers, and the weight matrices implicitly promotes the interaction of different channels. We modify **Figure 1(b)** to the weight matrix product of MLP layers in previous Transformer models to intuitively compare with the adjacency matrix constructed using Pearson correlation coefficient. The result of weight matrix product leads to a correlation matrix with chaotic data distribution, which may result in strongly correlated sequences not interacting effectively and weakly correlated sequences introducing noise to each other.
> 2. **Temporal dependencies.** Transformer is good at capturing the temporal dependencies of sequence data because of the use of self-attention mechanism with position encoding. However, classic position encoding methods such as fixed and learnable position encoding are first introduced in NLP tasks. These methods focus more on positional variance, not the strict temporal order, which is a key characteristic of time series data. This motivate us to find a better position encoding method to inject strict temporal order information to sequence tokens, and RNN is a good choice because of its sequence modeling capability. The ablation study results shown in **Table 3** can be the supporting evidence for the above design. The use of RNN-based position encoding reduces the MSE and MAE by 2.51% and 1.93% on average.
>
> > Q6: The authors mentioned that they used multi-layered RNNs, however in the appendix, it's said 1-layer RNN was used. Can you clarify the details of the RNN architecture?
>
> A6: We thank the reviewer for the detailed question. In the earlier version of the paper, we lack the sensitivity analysis on the number of RNN layers ($C$). Based on your comment, in the revised version, we conduct experiments on this hyperparameter and add the results in **appendix (A 5.1)**. We sum the result in the following table, for weather, electricity, ETTh1, ETTm1 and ILI, the model reaches optimal performance with $C=1$; for traffic, ETTh2 and ETTm2, the best RNN layer number is $C=2$.
> | Datasets | Weather | Traffic | Electricity | ETTh1 | ETTh2 | ETTm1 | ETTm2 | ILI |
> | :---: | :---: | :---: | :---: | :---: | :---: | :---: | :---: | :---: |
> | Optimal $C$ | 1 | 2 | 1 | 1 | 1 | 2 | 2 | 1 |
>
> > Q7: “To properly capture temporal dependencies, we consider using a multilayer RNN to encode the positions in the time series.” Why deep RNNs can properly capture temporal dependencies while Transformers can’t?
>
> A7: Thank you for the comment, the description in the earlier version may cause confusions, so we have modified it to “In order for the model to make use of the strict temporal order of the sequence, we consider using a multi-layer RNN to inject some positional contextual information into the sequence tokens” in **section 3.2.1**. Our viewpoint is that vanilla Transformer can indeed capture temporal dependencies, however, the position encoding methods (e.g. fixed or learnable position encoding) are initially designed for NLP tasks, where the sequence data used in nature language do not emphasize strict temporal order as much as time series data does. We hope to use the sequence modeling capability of RNN to supplement this temporal order information, so that the Transformer-based models can learn this characteristic of time series and obtain more reasonable temporal dependencies.

---

> ### Author Response · Authors · 2023-11-21
> **Response to Reviewer rJkP (Q1,Q2,Q3 and Q4)**
>
> > Q1: I am not sure what I am seeing in Figure 1(b), and I don’t understand how to interpret the authors' claim that cross-channel interaction is chaotic based on simply visualizing the weight matrices of the Transformer's dense layer (internal MLP).
>
> A1: We are thankful for your suggestion. We modify the corresponding explanations of **Figure 1(b)** to make it clear. In the revised version, we save the weight matrix parameters and calculated their matrix products, and get an equivalent correlation matrix. Values of each channel of a multi-variant time series will be passed through an embedding mapping projector, a series of feedforward networks of Encoders and Decoders, and a final mapping projector. These neural networks are usually some MLP layers, such as Linear layer and Conv1d layer. The weight matrix plays an important part in promoting interactions across different channels.
> Compared with the Pearson correlation coefficient matrix (or the correlation matrix obtained by any other algorithms), this result does not effectively promote the interaction between strong correlated sequences and prevent weak correlated sequences from interacting, either.
>
> > Q2: I am not sure I understand the authors' point about positional encoding not being able to represent temporal orders well. RNNs have their own problems, such as vanishing gradients when modeling long-term temporal dependencies. Are you suggesting that RNNs outperform Transformers in multivariate long-term time series forecasting (MLTSF)? -> Are the ablation results in Table 3 the experiments to back this claim? If that's the case, the performance difference between an RNN-based positional encoding (?) vs a learned positional embedding is almost 0.
>
> A2: Thank you for this thoughtful question. We agree with you that RNN has problems such as vanishing gradients facing long-term sequences, however, our motivation is to make use of the sequence modeling capability of RNN only to generate position encodings and we express this point in our **contributions** and **section 3.2.1**. The output results of the RNN layer are not directly used as the final feature embeddings, and this could alleviate the vanishing gradient problems.
> We make more detailed parameter adjustments to the experimental setup of the ablation study, such as changing the number of RNN layers $C$ to observe performance changes. We choose the best results and update **Table 3** in **section 4.2**. overall, RNN-based position encoding achieves 2.51% and 1.93% improvement in MSE and MAE metrics, respectively. We present the optimal results using RNN-based position encoding with layers 1, 2, and 3 on all the dataset in **appendix (A 5.1)**.
>
> > Q3: What exactly is the RNN-based position encoding method? In the caption for Figure 2, it says "The multi-layer RNN injects temporal order information." However, RNNs are not just injecting temporal order information as some sort of advanced positional encoding method; they can actually learn temporal dependencies. I am not sure if you are distinguishing between positional encoding and learning temporal representation.
>
> A3: Thank you for asking this question. Passing data into RNN can indeed extract temporal dependencies, however, this operation may cause information loss because of gradient explosion and vanishing when facing long-term sequence. In our model, RNN-based position encoding refers to the use of RNN structures for only generating position encodings with a feature-driven scheme. We hope to use its sequence modeling characteristics to generate position encoding with strict temporal order information, rather than directly output its results as temporal dependent features. By doing so, we can obtain enhanced position encodings compared to position encodings used by vanilla Transformer-based models and help the model to comprehend the strict temporal order of time series data.
>
> > Q4: Figure 2 (b) is hard to understand, at least explain the operator signs in the caption, arrows are not clear.
>
> A4: We apologize for the confusion. **Figure 2(b)** shows the module design based on multi-layer RNN position encoding, all the operator signs are in **Eq(1)**. We add explainations for **Figure 2(b)** in the revised version. The hidden state of the $j$-th RNN unit in $(c-1)$-th layer $r_{(i, j)}^{(c-1)}$ is used as inputs for RNN units at the same position in the next layer (the $j$-th RNN unit in the $c$-th layer) and the previous time step hidden state of RNN units at the next position in the same layer (the $(j+1)$-th RNN unit in the $(c-1)$-th layer). We update the arrows in **Figure 2(b)** and provide a more detailed explanation.

---

### Meta-Review · Area_Chair_xLtv · 2023-12-06

**Metareview:**

The reviewers raised multiple concerns, including the novelty representation of the RNN-based position encoding and GNN, some graph notations, etc. Although the author feedback has largely improved the manuscript, some concerns have not been resolved yet, which suggests a reject. The authors are suggested to include all the revisions and present a better future version of this paper.

**Justification For Why Not Higher Score:**

The concerns of the reviewers have not been resolved yet, which suggests a reject for this submission.

**Justification For Why Not Lower Score:**

n/a

---

### Decision · Program_Chairs · 2024-01-16

Reject